# Bacterial symbiont subpopulations have different roles in a deep-sea symbiosis

Tjorven Hinzke[1,2,3], Manuel Kleiner[4], Mareike Meister[5,6], Rabea Schlüter[7], Christian Hentschker[8], Jan Pané-Farré[9], Petra Hildebrandt[8], Horst Felbeck[10], Stefan M Sievert[11], Florian Bonn[12], Uwe Völker[8], Dörte Becher[5], Thomas Schweder[1,2], Stephanie Markert[1,2]*

[1]Institute of Pharmacy, University of Greifswald, Greifswald, Germany; [2]Institute of Marine Biotechnology, Greifswald, Germany; [3]Energy Bioengineering Group, University of Calgary, Calgary, Canada; [4]Department of Plant and Microbial Biology, North Carolina State University, Raleigh, United States; [5]Institute of Microbiology, University of Greifswald, Greifswald, Germany; [6]Leibniz Institute for Plasma Science and Technology, Greifswald, Germany; [7]Imaging Center of the Department of Biology, University of Greifswald, Greifswald, Germany; [8]Interfaculty Institute for Genetics and Functional Genomics, University Medicine Greifswald, Greifswald, Germany; [9]Center for Synthetic Microbiology (SYNMIKRO), Philipps-University Marburg, Marburg, Germany; [10]Scripps Institution of Oceanography, University of California San Diego, San Diego, United States; [11]Biology Department, Woods Hole Oceanographic Institution, Woods Hole, United States; [12]Institute of Biochemistry, University Hospital, Goethe University School of Medicine Frankfurt, Frankfurt, Germany

*For correspondence:
stephanie.markert@uni-greifswald.de

Competing interests: The authors declare that no competing interests exist.

**Abstract** The hydrothermal vent tubeworm *Riftia pachyptila* hosts a single 16S rRNA phylotype of intracellular sulfur-oxidizing symbionts, which vary considerably in cell morphology and exhibit a remarkable degree of physiological diversity and redundancy, even in the same host. To elucidate whether multiple metabolic routes are employed in the same cells or rather in distinct symbiont subpopulations, we enriched symbionts according to cell size by density gradient centrifugation. Metaproteomic analysis, microscopy, and flow cytometry strongly suggest that *Riftia* symbiont cells of different sizes represent metabolically dissimilar stages of a physiological differentiation process: While small symbionts actively divide and may establish cellular symbiont-host interaction, large symbionts apparently do not divide, but still replicate DNA, leading to DNA endoreduplication. Moreover, in large symbionts, carbon fixation and biomass production seem to be metabolic priorities. We propose that this division of labor between smaller and larger symbionts benefits the productivity of the symbiosis as a whole.

## Introduction

The chemoautotrophic gammaproteobacterium *Candidatus* Endoriftia persephone, sulfur-oxidizing endosymbiont of the deep-sea tubeworm *Riftia pachyptila* (here *Riftia*), provides all nutrition for its gutless host (*Cavanaugh et al., 1981*; *Felbeck, 1981*; *Hand, 1987*; *Robidart et al., 2008*). *Ca.* E. persephone (here Endoriftia) densely populates *Riftia*'s trophosome, a specialized organ in the worm's trunk, where the bacteria are housed intracellularly in host bacteriocytes (*Hand, 1987*). Here, fueled by the oxidation of reduced sulfur compounds (mainly sulfide), the symbionts fix $CO_2$ and produce considerable amounts of biomass, which constitutes the host's source of nutrition (*Cavanaugh et al., 1981*; *Felbeck et al., 1981*; *Childress et al., 1991*; *Goffredi et al., 1997a*;

*Girguis and Childress, 2006*). *Riftia* acquires these nutrients mainly by directly digesting parts of the symbiont population (*Hand, 1987*; *Boetius and Felbeck, 1995*; *Hinzke et al., 2019*). In addition, the symbionts possibly also actively secrete carbon compounds into the bacteriocytes (*Felbeck and Jarchow, 1998*). Thus provided, *Riftia* grows rapidly and reaches nearly 2 m in body length (*Jones, 1981*; *Lutz et al., 1994*; *McClain et al., 2015*), despite its lack of a digestive system. This enormous productivity is underpinned by a multitude of symbiosis-specific adaptations of both host and symbiont. These include host proteins involved in substrate transport from the hydrothermal vent environment to the symbionts, such as ATPases and carbonic anhydrases for $CO_2$ uptake and conversion (*Goffredi et al., 1997b*; *De Cian et al., 2003*), and specialized extracellular hemoglobins which simultaneously and reversibly bind sulfide and oxygen (*Childress et al., 1984*). Moreover, symbiont cells in the highly vascularized trophosome contain globules of elemental sulfur, an intermediate of sulfide oxidation (*Bright and Sorgo, 2003*; *Pflugfelder et al., 2005*). As trophosome sulfur content is positively correlated with sulfide concentrations in the surrounding water and decreases in the absence of external sulfide (*Childress et al., 1991*; *Robidart et al., 2011*; *Scott et al., 2012*), sulfur in the symbionts likely serves as an energy reserve for times of sulfide deprivation.

Although the symbiont population consists of a single 16S rRNA phylotype (comprising one dominating and several minor genotypes; *Polzin et al., 2019*), it was previously shown to exhibit remarkable metabolic versatility and redundancy: As demonstrated by proteomic analyses, symbionts from the same host animal expressed enzymes of two $CO_2$ fixation pathways, the Calvin cycle and the reverse tricarboxylic acid (rTCA) cycle, as well as enzymes for both glycogen generation and glycogen degradation (*Markert et al., 2007*; *Markert et al., 2011*; *Gardebrecht et al., 2012*; *Hinzke et al., 2019*). Moreover, proteins involved in utilization of hydrogen sulfide and thiosulfate as energy sources were expressed simultaneously by the same symbiont population; as were proteins for the use of nitrate and oxygen as electron acceptors (*Markert et al., 2011*). Based on these observations, we hypothesized that individual, metabolically distinct symbiont subpopulations in the trophosome may exist.

These presumptive subpopulations are likely congruent with symbionts of different cell sizes: Individual Endoriftia cells exhibit pronounced morphological diversity, ranging from small rods to small and large cocci in ultimate proximity to each other within the same host specimen (*Hand, 1987*; *Bright et al., 2000*; *Bright and Sorgo, 2003*). In individual trophosome lobules, which measure approximately 200–500 μm in diameter, the smallest, rod-shaped symbiont cells are located close to the central blood vessel, whereas toward the lobule periphery, symbionts gradually increase in size and become coccoid, before they are degraded in the outermost lobule zone. Only small Endoriftia cells and the host bacteriocytes in which they reside appear to undergo cell division, indicating that small and large symbionts belong to a common cell cycle (*Bright et al., 2000*; *Bright and Sorgo, 2003*). Previous microscopy-based studies indicated that small and large *Riftia* symbionts differ not only with regard to their frequency of cell division, but also with regard to carbon incorporation rates, amount of stored glycogen, and area of sulfur storage vesicles (*Bright et al., 2000*; *Sorgo et al., 2002*; *Bright and Sorgo, 2003*; *Pflugfelder et al., 2005*). This suggests that individual cell sizes may indeed have dissimilar metabolic properties.

In this study, we aimed to analyze and compare the metabolic profiles of individual *Riftia* symbiont subpopulations. Unlike previous molecular analyses that studied the uncultured *Riftia* symbiont's metabolic capabilities on the population level in a mixture of all cell sizes (e.g. *Markert et al., 2007*; *Markert et al., 2011*; *Gardebrecht et al., 2012*), precluding comparisons between putative subpopulations, we used a more sensitive approach. We enriched Endoriftia cells of different sizes by gradient centrifugation of trophosome tissue homogenate and subjected these enriched gradient fractions to separate metaproteomic analyses. Statistical evaluation using clustering and random forests allowed us to deduce cell size-dependent differences in protein abundance and metabolic functions. To assess whether these differences might be influenced by naturally occurring variations in energy supply, we conducted our analyses with symbionts from both sulfur-rich and sulfur-depleted trophosomes. Catalyzed reporter deposition-fluorescence in situ hybridization (CARD-FISH), transmission electron microscopy (TEM), hybridization chain reaction (HCR)-FISH analyses, and flow cytometry complemented these experiments. Our results suggest a division of labor between different developmental stages of the symbiont, which leads to remarkable physiological heterogeneity within the symbiont population.

## Results

### Enrichment of individual symbiont cell sizes by gradient centrifugation

Our rate-zonal gradient centrifugation approach allowed us to enrich distinct symbiont cell sizes from *Riftia* trophosome tissue. Based on CARD-FISH microscopy, we defined four size ranges (*Figure 1*): very small symbiont cells ($\geq$2.0 –<3.9 µm diameter), small ($\geq$3.9 –<5.3 µm), medium ($\geq$5.3 –<6.8 µm), and large symbiont cells ($\geq$6.8–20.0 µm; see also Appendix section A). For subsequent comparative metaproteomic analyses, we chose those gradient fractions that were most enriched in one of these cell size ranges. In the following, these four gradient fractions are referred to as XS (containing the highest percentage of very small symbiont cells) to L (containing the highest percentage of large cells). The enrichment procedure was highly reproducible, particularly for symbionts isolated from sulfur-rich trophosome tissue (*Figure 1*).

### Symbiont DNA quantification

Flow cytometry and fluorescence-activated cell sorting (FACS) indicated that DNA content in large *Riftia* symbionts is up to 10-fold higher compared to small symbionts. To identify distinguishable bacterial cell populations, we examined Syto9-stained cells in *Riftia* trophosome homogenate and in gradient fractions from the upper and lower parts of the gradient (enriched in smaller and larger symbionts, respectively) with regard to their light scattering properties. Forward scatter (FSC) and side scatter (SSC) usually correlate with cell size and cell granularity, respectively (*Bouvier et al., 2001*; *Tracy et al., 2010*). Amongst a number of particle groups with different properties (see also Appendix section C), we found two populations, 1 and 2, which were abundantly detected in non-enriched trophosome homogenate, but showed very dissimilar frequencies in fractions enriched in larger or smaller symbionts (*Figure 2*, *Figure 2—figure supplement 1*): While population 1, which exhibited relatively lower FSC and SSC signals (indicative of smaller cell size and lower cell complexity), was highly abundant in fractions enriched in smaller symbionts, this population was notably less prominent in fractions enriched in larger symbionts. Simultaneously, population 2, which gave higher FSC and SSC signals (indicative of larger cell size and higher complexity), was highly abundant in fractions enriched in larger symbionts but nearly absent in gradient fractions enriched in smaller symbionts. This suggests that populations 1 and 2 consist of smaller and larger symbionts, respectively. This assumption was verified by FACS-separation of both populations from trophosome homogenate, and examination of the sorted cell suspensions by fluorescence microscopy along with unsorted enriched gradient fractions and homogenate samples for reference (*Figure 2—figure supplement 1*). For quantification of DNA in smaller and larger symbionts, we compared median fluorescence intensities (MFI) per particle between populations 1 and 2 in non-enriched homogenate and in enriched gradient fractions. In all sample types, MFI per particle was notably lower in population 1 (between 186 and 1994 relative fluorescence units, rfu) than in population 2 (2,712–10,723 rfu). On average, MFI was 9.7-fold higher in population 2 than in population 1 (*Supplementary file 6*).

### Protein identifications and relative protein abundance

We identified a total of 1946 symbiont proteins across all sample types, including the four gradient fractions XS – L and non-enriched homogenate from both sulfur-rich and sulfur-depleted *Riftia* specimens (*Appendix 1—table 2*). Our sample fractionation by gradient centrifugation thus facilitated detection of around 60% of the symbiont's theoretical proteome, which encompasses 3182 proteins in PRJNA60889, and yielded substantially higher symbiont protein identification rates than non-enriched trophosome homogenate samples alone (1223 total symbiont protein identifications). After stringent filtering and normalization, a subset of 1212 symbiont proteins from gradient fractions XS – L was included in statistical analysis using abundance profile clustering and random forests (*Supplementary file 2*). A total of 465 proteins showed significant differences in relative abundance (*Appendix 1—figure 1*; note that the term 'significant' denominates trends that were consistent across all replicates in the context of our statistical approach). In *Figure 3* and *Supplementary file 2*, proteins that showed such significant changes in relative abundance are marked with asterisks. Of all proteins with significant abundance changes, 56% (261 proteins)

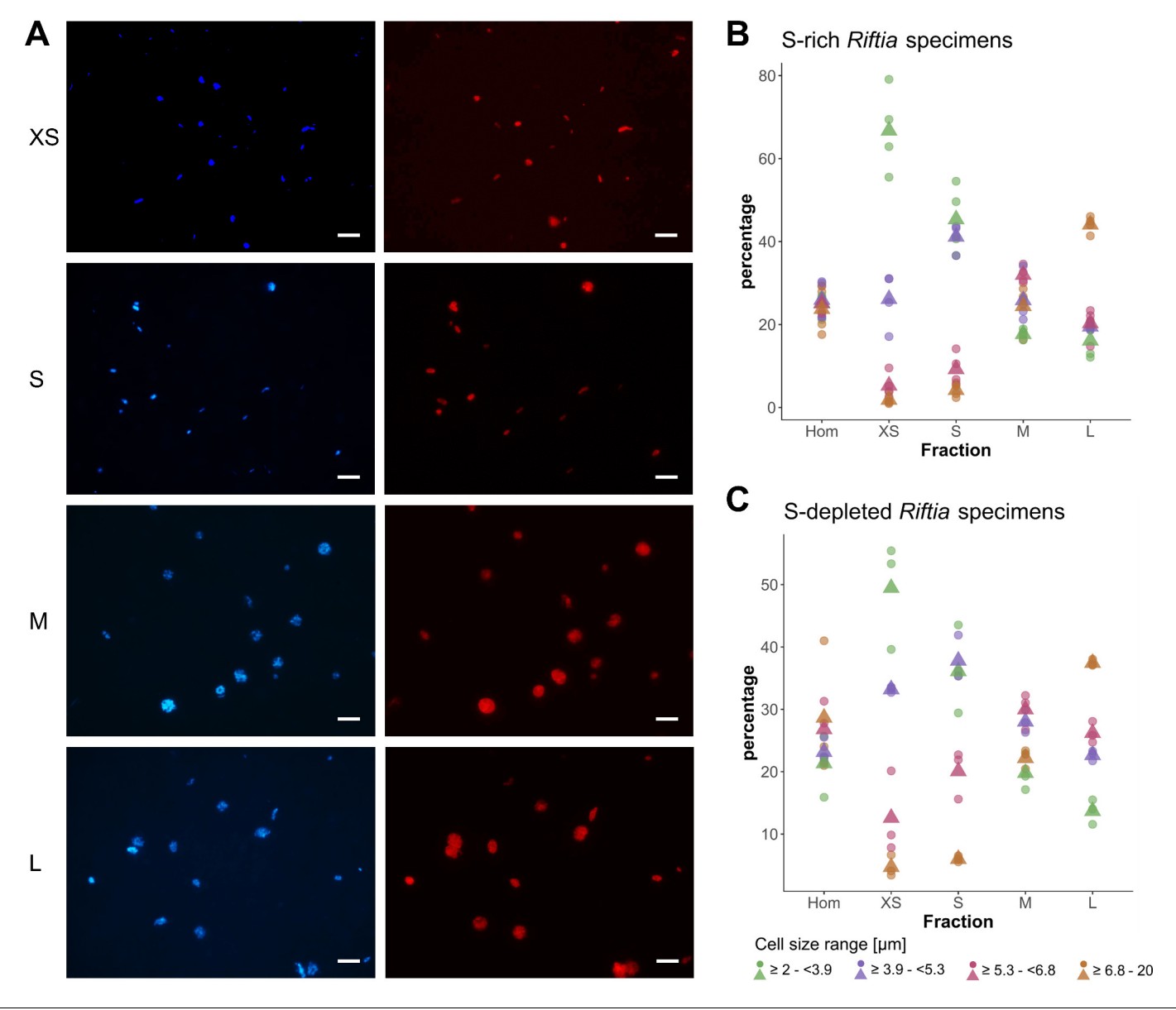

**Figure 1.** CARD-FISH images of *Riftia* symbiont cells after density gradient centrifugation. (A) Catalyzed reporter deposition-fluorescence in situ hybridization (CARD-FISH) images of *Riftia* symbiont cells after density gradient centrifugation of trophosome homogenate. After the enrichment procedure, small bacterial cells had accumulated in the upper, less dense gradient fractions (top), whereas larger symbionts were enriched in the lower, denser fractions (bottom). Left: DAPI staining, right: 16S rRNA signal. For better visibility, brightness and contrast were adjusted in all images. Between 300 and 1300 individual cells were measured per filter (average: 590). (B and C) Symbiont cell size distributions in individual gradient fractions. While all cell size groups were roughly equally abundant in non-enriched trophosome homogenate (Hom), fraction XS had the highest percentage of symbiont cells in the size range 2.0 µm - 3.9 µm, fraction S contained most symbiont cells of 3.9 µm – 5.3 µm, etc. Gradient centrifugation was performed using four biological replicates (n = 4) of sulfur-rich trophosome tissue (B) and three biological replicates (n = 3) of sulfur-depleted trophosome tissue (C). For an overview of which gradient fractions were chosen as fractions XS, S, M, and L in all samples see ***Supplementary file 1***. Dots: individual % values, triangles: average % values. Please note the different scaling of the y axes in B and C.

followed a clear, continuous abundance trend from fraction XS to L or vice versa, that is, protein abundance increased or decreased with increasing symbiont cell size (***Supplementary file 2***).

In our comparison of energy-rich (S-rich) and energy-depleted (S-depleted) gradient fractions, very few proteins were detected exclusively in sulfur-rich samples (61 of 1212 proteins) or only in sulfur-depleted samples (77 proteins). The majority of symbiont proteins showed very similar

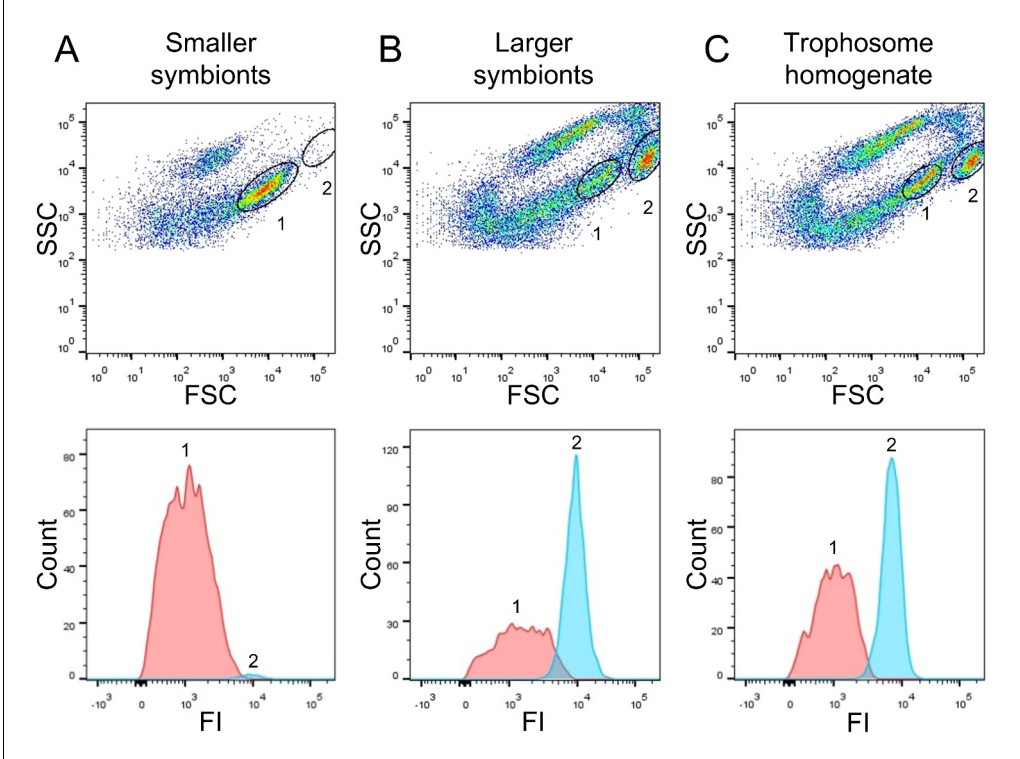

**Figure 2.** Flow-cytometry-based DNA quantification of *Riftia* symbionts. (**A**) Dot plot of forward scatter (FSC) and side scatter (SSC), and histogram with fluorescence signal counts and fluorescence intensity (FI) per particle of a gradient fraction enriched in smaller symbionts. (**B**) Gradient fraction enriched in larger symbionts. While cell population 1 was more prominent in (**A**), population 2 was almost exclusively detected in (**B**), and both populations were present in non-enriched trophosome homogenate (**C**), indicating that population 1 corresponds to smaller symbionts, whereas population 2 corresponds to larger symbiont cells. Cells were stained with Syto9 and median fluorescence intensity (MFI) per particle at wave length 530/30 nm was used as a measure of cellular DNA content (see *Figure 2—figure supplement 1*, Methods, and *Supplementary file 6* for more details). This analysis was based on two *Riftia* specimens with medium sulfur content.
The online version of this article includes the following figure supplement(s) for figure 2:

**Figure supplement 1.** Fluorescence microscopy and fluorescence-activated cell sorting (FACS) of *Riftia* symbiont cells.

---

abundance trends in sulfur-rich and sulfur-depleted samples. The following results will therefore not discriminate between both sample types and subsequent figures will focus on S-rich samples (unless otherwise stated). For a discussion of specific differences observed between symbionts from energy-rich and energy-starved trophosome tissue, see Appendix section B.

## Symbiont protein functions
### Cell cycle, DNA topology, replication and repair
Proteins involved in the bacterial cell cycle and in DNA topology, -replication and -repair were differentially expressed across fractions XS to L (*Figure 3*, *Supplementary file 3a*). While the cell division protein FtsZ, as well as DNA gyrase and DNA-binding proteins decreased significantly in abundance from fraction XS to L, abundance of other cell-division-related proteins (e.g. FtsE, MreB, division inhibitor SlmA), and of proteins involved in DNA replication (e.g. DNA ligase, DNA polymerase) and repair (e.g. UvrAB) increased. Interestingly, FtsZ abundance was very low in S-depleted fractions, so that it was excluded from statistical analysis in these samples (see Appendix section B).

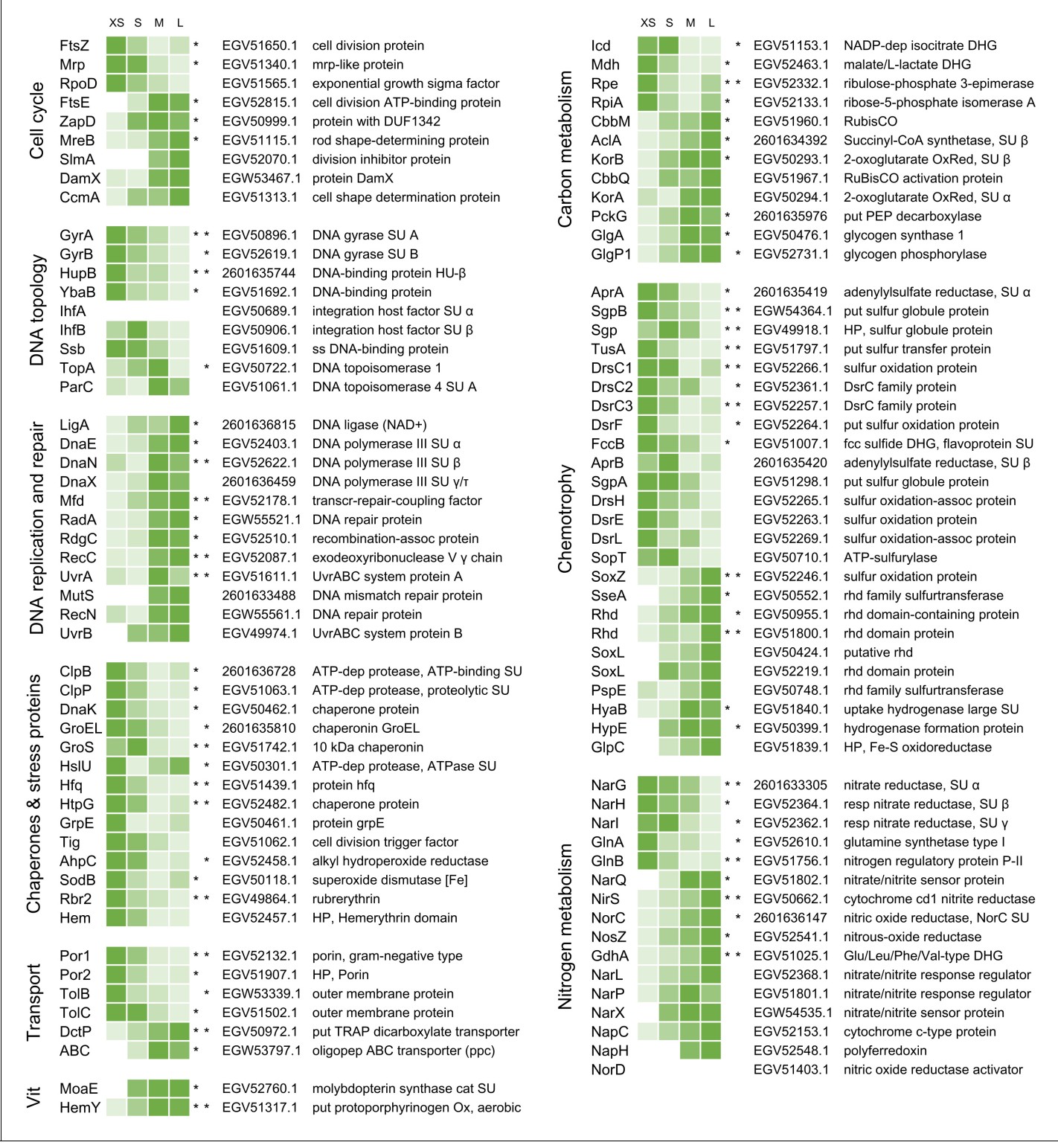

**Figure 3.** Abundance trends of selected Endoriftia proteins of various functions in the four fractions XS to L. Trends are indicated by color shades from light green (lowest protein abundance across all four fractions) to dark green (highest abundance across all four fractions; note that colors do not allow comparison of protein abundance between proteins). Abundance values in the heat map are based on statistical evaluation of four biological replicates with sulfur-rich trophosomes (for abundance trends of sulfur-depleted samples refer to *Figure 3—figure supplement 1* and Appendix section B). Proteins marked with asterisks show statistically significant trends, that is, differences that are consistent across all replicates in S-rich (left asterisk) or S-depleted specimens (right asterisk), or both (two asterisks). White cells indicate that this protein was not detected in this sample or too low abundant

Figure 3 continued

to be included in statistical analyses. For an overview of all identified symbiont proteins and their relative abundances and for a summary of protein abundance trends sorted by metabolic category see *Supplementary files 2* and *3*, respectively. Accession numbers refer to NCBI/JGI entries. SU: subunit, DUF: domain of unknown function, ss: single-stranded, transcr: transcription, assoc: associated, dep: dependent, HP: hypothetical protein, put: putative, oligopep: oligopeptide, ppc: periplasmic component, DHG: dehydrogenase: RubisCO: ribulose-1.5-bisphosphate carboxylase/oxygenase, Ox: oxidase, OxRed: oxidoreductase, PEP: phosphoenolpyruvate, fcc: flavocytochrome c, rhd: rhodanese, resp: respiratory, cat: catalytic, Vit: vitamin and cofactor metabolism.

The online version of this article includes the following figure supplement(s) for figure 3:

**Figure supplement 1.** Abundance trends of selected Endoriftia proteins of various functions in the four fractions XS to L in sulfur-rich (S-rich) and sulfur-depleted (S-depl) *Riftia* specimens.

## Chaperones and stress proteins

Many chaperones and other proteins involved in protein folding, as well as oxidative stress-related proteins were detected with significantly decreasing abundance from fraction XS to L, including (amongst others) the proteases ClpB, ClpP, GroEL, the abundant alkyl hydroperoxide reductase AhpC, superoxide dismutase SodB, and rubrerythrin (*Figure 3*, *Supplementary file 3b*).

## Transport

Outer membrane proteins such as two porins and TolBC showed significant abundance differences between the fractions, with highest relative abundance in fraction XS and lowest abundance in fractions L. Porin EGV52132.1 (Por1) was the most abundant symbiont protein throughout all sample types (*Figure 3*, *Supplementary file 3c*). On the other hand, all five detected tripartite ATP-independent periplasmic (TRAP) transporter subunits and 10 out of 13 ABC transporter components were relatively more abundant in fraction L (see also Appendix section D).

## Central metabolism

*Carbon metabolism:* Several tricarboxylic acid (TCA) cycle enzymes (e.g. Icd, Mdh), as well as enzymes of the pentose phosphate pathway (e.g. Rpe, RpiA) were detected with *decreasing* abundances from fraction XS to L (*Figure 3*, *Supplementary file 3d*). In contrast, the key enzymes of the two $CO_2$-fixing pathways, Calvin cycle (ribulose-1.5-bisphosphate carboxylase/oxygenase, RubisCO, CbbM) and rTCA cycle (ATP-citrate lyase, AclA; oxoglutarate oxidoreductase, KorAB), as well as most of the gluconeogenesis-related (e.g. PckG), and glycogen metabolism-related enzymes (e.g. GlgA, GlgP) increased in abundance from fraction XS to L. Relative stable isotope fingerprint (SIF) values, that is, relative $\delta^{13}C$ values, generally also increased from fraction XS to L (see Appendix sections B and D for details).

*Chemotrophy*: Many sulfide oxidation-specific proteins, including both subunits of the abundant key enzyme adenylylsulfate reductase AprAB, as well as proteins involved in sulfur storage (sulfur globule proteins) had their highest abundance in fraction XS or S and their lowest abundance in fraction M or L (*Figure 3*, *Supplementary file 3e*, *Appendix 1—figure 4*). In contrast, thiosulfate oxidation-related proteins like SoxZ, SoxL, and other rhodanese-like proteins were detected with significantly increasing abundance from fraction XS to fraction L. Four additional Sox proteins, that is, SoxA, SoxB, SoxW, and SoxY, which were detected at very low abundances across the sample types (and were therefore excluded from statistical analysis), were identified in fraction M and L, but were completely absent from fraction XS (*Supplementary file 2*). Three proteins involved in energy generation by hydrogen oxidation, HyaB, HypE and GlpC, were also detected with increasing abundance from fraction XS to fraction L.

*Nitrogen metabolism:* Relative abundance of all three respiratory membrane-bound nitrate reductase subunits, NarGHI, decreased significantly from fraction XS to L, as did abundance of glutamine synthetase GlnA (*Figure 3*, *Supplementary file 3f*, Appendix section E). On the other hand, various other denitrification-related proteins (such as nitrite reductase NirS, nitrous oxide reductase NosZ, and nitrate/nitrite signal transduction systems) and glutamate dehydrogenase GdhA showed relatively higher abundances in fraction L (or M) than in fraction XS. The same trend was observed for the periplasmic nitrate reductase components NapC and NapH. Moreover, NapG, another NapH copy, the nitric oxide reductase subunit NorB, nitric oxide reductase activation protein NorQ, and

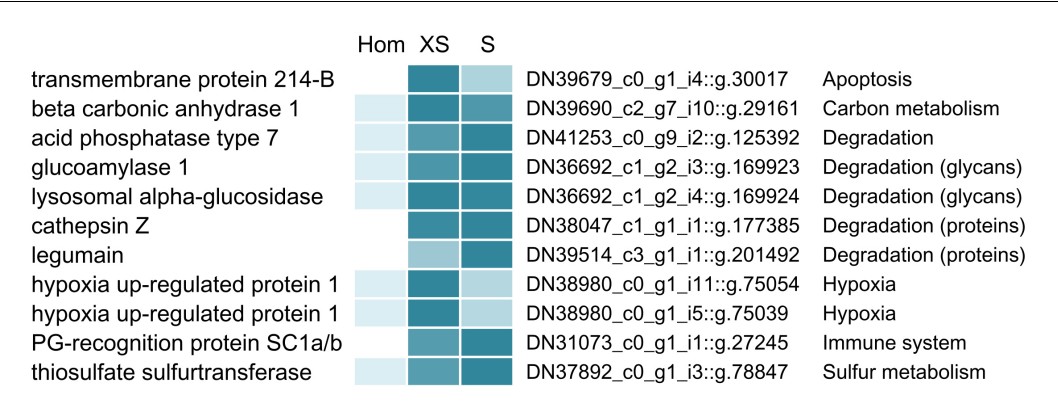

**Figure 4.** Selected *Riftia* host proteins with significantly higher relative abundance in the symbiont-enriched fractions XS and S compared to the non-enriched trophosome tissue homogenate (Hom). Relative abundance trends are indicated by color shades from light blue (lowest protein abundance across the three sample types) to dark blue (highest abundance), based on mean values from four biological replicates with sulfur-rich trophosome. (Note that colors do not allow comparison of protein abundance between proteins). Accession numbers refer to the combined host and symbiont database used for protein identification in this study (see Materials and methods). For a complete list of host proteins with significantly higher abundance in fractions XS and S (compared to Hom) see *Supplementary file 4*. This comparison includes only the symbiont-enriched fractions XS and S, but not fractions M and L, because these latter fractions were more likely to be contaminated by non-symbiosis-specific host proteins from host tissue fragments pelleted during centrifugation. PG: peptidoglycan.

the putative assimilatory nitrite reductase subunit NirB, whose overall abundances were too low to include them in statistical analysis, were only detected in fraction M and/or L.

### Other categories

Fifty (75%) of the 67 proteins involved in cofactor- and vitamin synthesis in S-rich samples had their highest abundance in fraction M or L (*Figure 3*, *Supplementary file 3g*). Also, of the 33 identified tRNA ligases and tRNA synthetases, 25 (75%) were most abundant in fraction M or L (in S-rich samples, *Supplementary file 3h*).

## Potentially symbiosis-specific host proteins

Our density gradient fractionation procedure allowed not only for the identification of symbiont proteins with differential abundance across different Endoriftia size ranges, but also enabled us to single out host proteins that are potentially involved in direct interactions with the symbionts. As host proteins that are attached to the symbionts are pulled down with the symbiont cells during gradient centrifugation, these proteins should be significantly more abundant in symbiont-enriched fractions compared to the non-enriched trophosome homogenate (*Figure 4*, *Supplementary file 4*). Besides many ribosomal and mitochondrial host proteins, which were also enriched, putatively symbiont-associated host proteins included the host's peptidoglycan-recognition protein SC1a/b, beta carbonic anhydrase 1, digestive proteins involved in protein- and carbohydrate degradation (for example, acid phosphatase, digestive proteases, and glycan degradation enzymes), as well as hypoxia up-regulated proteins, a thiosulfate sulfurtransferase, and transmembrane protein 214-B.

## Discussion

### Symbiont growth and differentiation

Cell division plays a more prominent role in small symbionts

As indicated by the significant decrease in abundance of the cell division key protein FtsZ from fraction XS to fraction L, small Endoriftia are more engaged in cell division than larger symbionts. In accordance with the microscopy-based hypothesis of *Bright and Sorgo, 2003*, the smallest

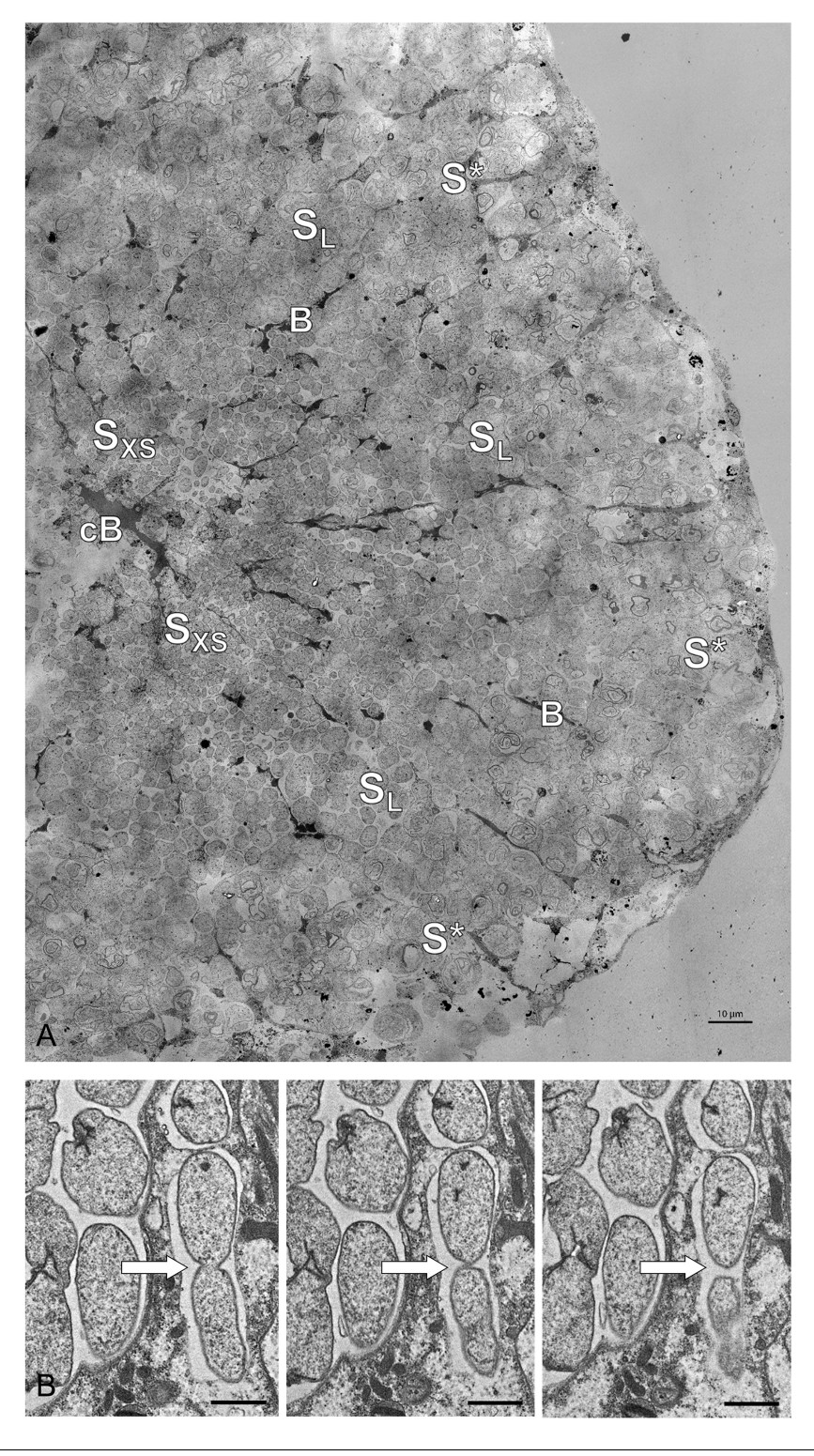

**Figure 5.** Electron microscopy of *Riftia* trophosome tissue. (**A**) Electron micrograph of a cross-section through a *Riftia* trophosome lobule. Surrounding an efferent central blood vessel (cB), small symbiont cells ($S_{XS}$) are visible in bacteriocytes (i.e. symbiont-containing host cells) in the central lobule zone. Symbiont cell size increases toward the periphery of the lobule ($S_L$: large symbiont cells). In the outermost bacteriocytes, symbiont cells are digested by host enzymes ($S^*$). Bacteriocytes are interspersed with smaller blood vessels (B), which facilitate blood flow from the lobule periphery to the lobule center (*Felbeck and Turner, 1995*). The image was assembled from 50 individual transmission electron micrographs of a trophosome section from a *Riftia* specimen with sulfur-depleted trophosome. The full resolution image is available as *Figure 5—figure supplement 1*. Contrast and brightness

*Figure 5 continued on next page*

*Figure 5 continued*

were adapted. (B) Cell division in small *Riftia* symbionts in the trophosome lobule center of a *Riftia* specimen with sulfur-rich trophosome. All micrographs show the same dividing Endoriftia cell in three subsequent tissue sections, revealing that both daughter cells are still connected, but are about to be separated (arrow). Scale bar: 1 µm. Despite thorough screening, we did not observe cell division in large Endoriftia cells in any of the TEM sections. This corroborates the idea that small and large symbiont subpopulations are developmental stages of the same Endoriftia strain.

The online version of this article includes the following figure supplement(s) for figure 5:

**Figure supplement 1.** High-resolution TEM image of the trophosome lobule section shown in *Figure 5*.

symbionts, which are in situ localized in the trophosome lobule center (*Figure 5*), thus apparently function as 'stem cells' of the symbiont population. During cell division, FtsZ forms the Z ring, to which the other division-related proteins are successively recruited (reviewed in *Weiss, 2004*). Cell size and cell division therefore likely depend on the amount of FtsZ available (*Chien et al., 2012*). This correlation is, for example, also reflected by a decrease of FtsZ concentration during differentiation of vegetative cells into non-dividing larger heterocysts in the cyanobacterium *Anabaena* (*Klint et al., 2007*). (Although originally referring to eukaryotic cells, the term 'differentiation' is also applied to bacteria with strongly varying morphologies, such as *Rhizobia* in legume root nodules and heterocyst-forming cyanobacteria. We will therefore also use it in the following to describe the development from small to large Endoriftia in this study.)

Interestingly, while FtsZ abundance decreased across fractions, many other proteins which interact with FtsZ during cell division were detected with increasing abundance from fraction XS to L. This indicates that these proteins are also involved in processes other than cell division, for example, in determining cell shape and stabilization. ZapD, for example, is involved in FtsZ filament organization, and its overexpression leads to cell filamentation (*Durand-Heredia et al., 2012*). DamX overexpression, too, was observed to induce filamentation in *E. coli* (*Lyngstadaas et al., 1995*), while overexpression of the cell shape determination protein CcmA in *E. coli* and *P. mirabilis* lead to enlarged, ellipsoidal cells (*Hay et al., 1999*), and FtsEX is required for cell elongation rather than cell division in *B. subtilis* (*Domínguez-Cuevas et al., 2013*). The actin homolog MreB is pivotal for rod-shape formation in bacteria and for cell stiffness in *E. coli*, could negatively regulate cell division, and participates in chromosome segregation (*Wachi and Matsuhashi, 1989*; *Kruse et al., 2006*; *Wang et al., 2010*, reviewed in *Reimold et al., 2013*). In large Endoriftia, these proteins might therefore be involved in stabilizing growing symbiont cells. SlmA, which was only detected in fractions M and L in our study, was shown to disassemble FtsZ polymers, thus acting as a cell division inhibitor (*Cho et al., 2011*), which supports the idea of relatively less cell division in large *Riftia* symbionts. Although Endoriftia's major cell division protein FtsZ was notably (1.75x) less abundant in fraction L (compared to fraction XS), it was not completely absent. This may point to additional FtsZ functions, besides cell division (as also suggested for *Anabeana* [*Klint et al., 2007*] and *E. coli* [*Thanedar and Margolin, 2004*]).

## Large symbionts have more genome copies and less compact chromosomes

Endoriftia's development into large, non-dividing (but still replicating) cells leads to endoreduplication cycles and an increase in genome copy number, as indicated by our flow cytometry analysis (*Figure 2*, *Figure 2—figure supplement 1*, *Supplementary file 6*). This observation is also in agreement with earlier findings of *Bright and Sorgo, 2003*, who noted more than one chromatin strand-containing area in large coccoid *Riftia* symbiont cells in electron microscopy images, whereas small rods and cocci featured only one chromatin strand area. The idea of endoreduplication in larger *Riftia* symbionts is additionally supported by the observation that large symbiont cells, which apparently divide less frequently than smaller cells (see above), still actively replicate DNA, as indicated by high abundances of DNA ligase and DNA polymerase III in fraction L. The observed decreasing abundance of DNA gyrase GyrAB with increasing cell size additionally corroborates this idea, as type II topoisomerases such as gyrase are not only involved in supercoiling and initiation of DNA replication (*Levine et al., 1998*; *Nöllmann et al., 2007*), but also essential for decatenation of newly replicated chromosomes in bacteria (*Steck and Drlica, 1984*; *Guha et al., 2018*). Moreover, inhibition of topoisomerase II in eukaryotes leads to endoreduplication and polyploidy (*Cortés and*

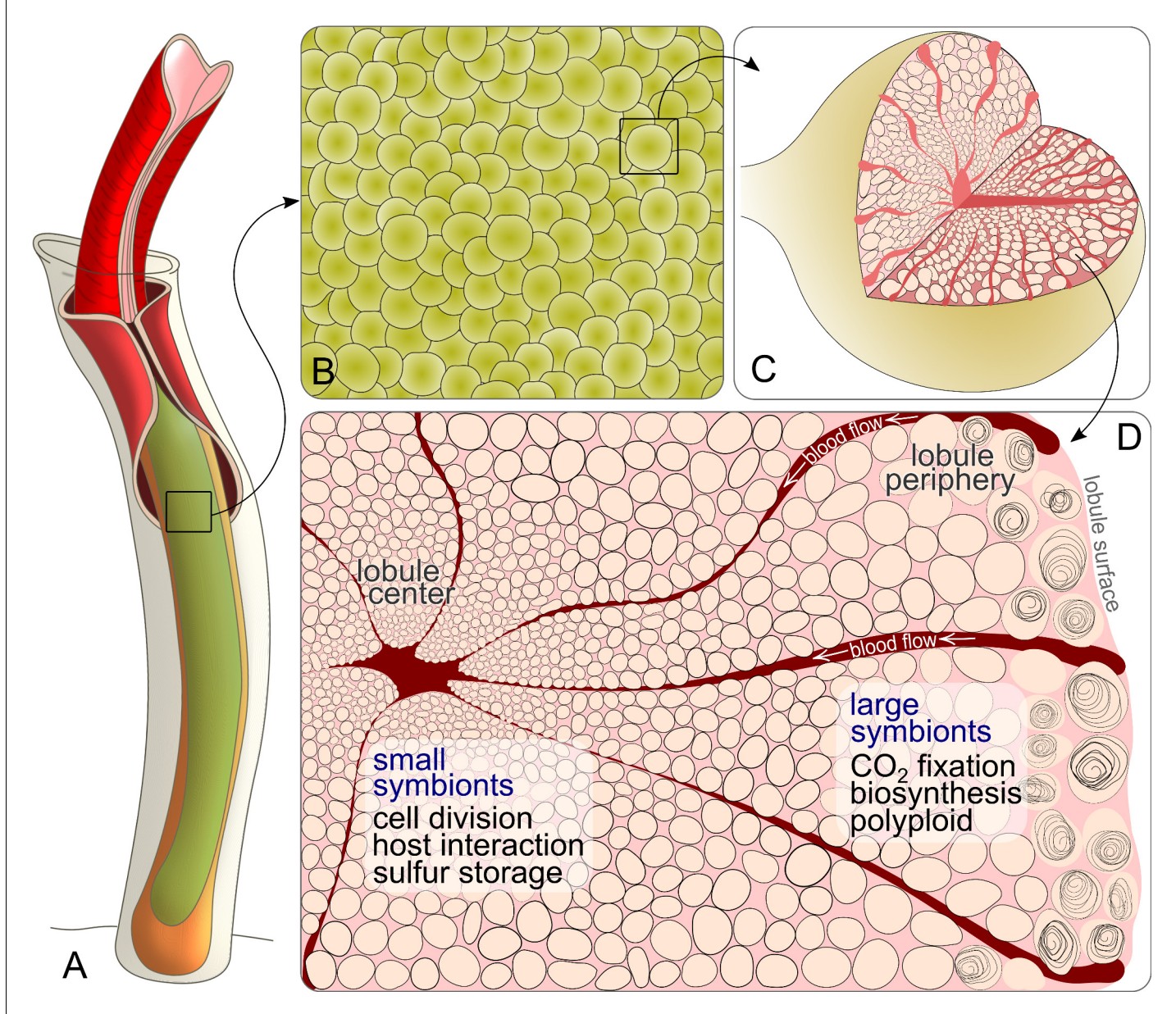

**Figure 6.** Schematic drawing of *Riftia* symbiont cells inside the trophosome. (**A**) An adult tubeworm reaches 2 m in body length. Its symbiont-containing organ, the trophosome (green), fills most of the body cavity (coelomic cavity), and is immersed in coelomic (non-vascular) blood (not shown). (**B**) Close-up of the lobular trophosome tissue. (**C**) Single lobule (200–500 µm in diameter) with interior blood vessels (blood-filled spaces) and symbiont cells visible. (**D**) Cross-section through a trophosome lobule (similar to that in *Figure 5A*) with small symbiont cells located in the center around an efferent blood vessel. Symbiont cell size increases toward the lobule periphery, where the largest symbionts are digested by the host (curls). Blood flow from lobule periphery to lobule center may cause gradients in nutrient availability. Based on the results of this study, the most striking characteristics of small and large symbionts, which determine their respective roles in the symbiosis, are indicated. Note that this is a simplified illustration, in which the various kinds of host cells (including their membranes, nuclei and organelles), as well as blood vessels or gonads that line the surface of the trophosome were omitted for clarity's sake. (Illustrations based on drawings, TEM images and descriptions by *van der Land and Nørrevang, 1977*; *Felbeck and Turner, 1995*; *Bright and Sorgo, 2003*).

*Pastor, 2003*; *Cortés et al., 2003*). Polyploidy in thiotrophic symbionts was also observed in the lucinid bivalve *Codakia orbicularis*, where larger symbiont cells contained more than four genome copies, whereas smaller cells had only one genome copy (*Caro et al., 2007*), and in ectosymbionts of *Eubostrichus* nematodes, in which up to 16 nucleoids per large symbiont cell were reported (*Polz et al., 1992*; *Pende et al., 2014*). Moreover, also terminally differentiating *Rhizobia* undergo

endoreduplication cycles (*Mergaert et al., 2006*), and high genome copy numbers have been reported for various bacterial insect symbionts, for example, of aphids, cockroaches, and sharp-shooters (*Komaki and Ishikawa, 2000*; *López-Sánchez et al., 2008*; *Woyke et al., 2010*), suggesting that polyploidy is common in symbiotic bacteria. Possibly, enlarged polyploid cells might increase the metabolic activity and/or fitness of the Endoriftia cells. In *E. coli*, an *mreB* point mutation led to increased cell size, which gave the cells a measurable fitness advantage in presence of certain carbon sources (*Monds et al., 2014*). Moreover, polyploidy was suggested to provide evolutionary advantages like a low mutation rate and resistance toward DNA-damaging conditions in haloarchaea (*Zerulla and Soppa, 2014*). In plants, endoreduplication is common and might increase transcription and metabolic activity of the cells (*Kondorosi and Kondorosi, 2004*), leading to enhanced productivity (*Sattler et al., 2016*). More generally, in symbiotic associations, where the bacteria are stably and sufficiently provided with carbon and energy sources, the advantages of polyploidy might be greater than the associated costs (*Angert, 2012*).

Higher genome copy numbers in large *Riftia* symbionts seem to be accompanied by a lower degree of DNA condensation, compared to small Endoriftia, as indicated by notably lower abundances of the histone-like DNA-binding proteins HU (HupB) and integration host factor (IHF, IhfAB), and of DNA gyrase GyrAB in fraction L, compared to XS. Bacterial histone-like DNA-binding proteins like HU and IHF structure the chromosome and modulate the degree of supercoiling (reviewed in *Dorman and Deighan, 2003*). In *E. coli*, absence of HU leads to unfolding of the chromosome and to cell filamentation (*Dri et al., 1991*), and unspecific DNA-binding by IHF was shown to contribute to DNA compaction (*Ali et al., 2001*). Moreover, bacterial DNA gyrase was also suggested to be involved in nucleoid compaction in *E. coli* (*Stuger et al., 2002*). Co-occurrence of endoreduplication and decondensated DNA is also known in plant cells (*Kondorosi and Kondorosi, 2004*). As decondensation occurs in actively transcribed DNA regions (*Wang et al., 2015*), it might facilitate protein synthesis and metabolic activity in large Endoriftia.

Since DNA condensation may function as a DNA protection mechanism (*Ohniwa et al., 2006*; *Mukherjee et al., 2008*; *Yoshikawa et al., 2008*; *Takata et al., 2013*), less condensed DNA might be more prone to various kinds of damage and require the enhanced expression of DNA repair mechanisms. This would explain the observed higher abundance of several DNA repair proteins in fraction L, which was enriched in larger, older symbiont cells with (presumably) larger quantities of less condensed DNA, compared to the smaller symbiont cells. RadA, RdgC, RecCN, UvrAB, and Mfd, which are known to be involved in DNA recombination and repair in many bacteria (*Kowalczykowski, 2000*; *Beam et al., 2002*; *Tessmer et al., 2005*; *Drees et al., 2006*; *Truglio et al., 2006*; *Deaconescu et al., 2007*), may compensate for this elevated vulnerability. In eukaryotes, chromatin decondensation was shown to facilitate access of the DNA damage response to double-strand breaks, thus allowing for more efficient repair (*Murga et al., 2007*).

## Small symbionts may be exposed to elevated stress levels

Small symbionts might experience cell-division-related or host-induced stress in the early phase of their cell cycle, as indicated by elevated levels of symbiont chaperones and stress response proteins, as well as of reactive oxygen species (ROS) scavengers in fraction XS. This is in line with observations in *Caulobacter crescendus*, where the DnaK-DnaJ and GroEL-GroES systems are crucial for cell division (*Susin et al., 2006*), and in *E. coli*, where the protease ClpXP and the RNA chaperone Hfq are probably involved in cell division as well (*Camberg et al., 2009*; *Zambrano et al., 2009*). Interestingly, like the putative Endoriftia 'stem cells', eukaryotic embryonic stem cells also feature high levels of chaperone expression and stress tolerance (*Prinsloo et al., 2009*). Although the reason for this congruence is yet unknown, one might speculate that cell-division-related processes require elevated levels of chaperones and stress proteins, for example, to ensure correct assembly of all parts of the division machinery or (in case of intracellular bacteria) to counteract some sort of yet to be determined host-induced stress.

Possibly, such host-induced stress may also involve the production of ROS in symbiont-containing bacteriocytes, similar to animal and plant hosts, which generate ROS to defend themselves against pathogenic bacteria (*Heath, 2000*; *Lynch and Kuramitsu, 2000*; *D'Haeze and Holsters, 2004*). Small symbionts, which are relatively loosely packed in their host cell vesicles (*Figure 5A*) and have a comparatively high surface-to-volume ratio, might be particularly exposed to this presumptive ROS

stress, while larger symbionts, which are more tightly packed, may face lower ROS levels. This would explain the observed higher abundance of the ROS scavengers rubrerythrin (Rbr2), superoxide dismutase (SodB), and alkylhydroperoxide reductase (AhpC) in small symbionts. In line with this assumption, a superoxide dismutase and also the chaperones ClpB, HtpG, and DnaK were suggested to be involved in ROS protection in *Serratia symbiotica* (*Renoz et al., 2017*), and ClpB protease expression has been shown to increase during oxidative stress in the intracellular pathogen *Francisella tularensis* (*Twine et al., 2006*).

Interestingly, we found no indications of a strong bacterial stress response in fraction L, indicating that imminent digestion by the host poses no particular stress to the large symbionts. Possibly, bacterial degradation happens too fast to elicit a stress response, or a digestion-related stress response is suppressed during symbiosis, either by the symbionts themselves or by the host via a yet to be determined mechanism.

## Host-microbe interactions may be particularly important in small Endoriftia

Abundant Endoriftia membrane proteins might play a key role in host interaction in small symbionts. Particularly, the high and differential abundance of porin Sym EGV52132.1, the most abundant symbiont protein in all fractions, which was nearly three times more abundant in fraction XS (11.7%orgNSAF) than in fraction L (4.0%orgNSAF), suggests that this protein may be of varying relative importance throughout the symbiont's differentiation process. Porins are water-filled channels in the outer membrane, through which small hydrophilic molecules can diffuse (*Fernández and Hancock, 2012*). In the oyster pathogen *Vibrio splendidus*, the porin OmpU serves as adhesin or invasin and is involved in recognition by the host cell (*Duperthuy et al., 2011*), while in *Neisseria gonorrhoeae*, a porin inhibits phagocytosis by human immune cells (*Mosleh et al., 1998*; *Lorenzen et al., 2000*). Interestingly, the phagocytosis-inhibiting action of *N. gonorrhoeae* porin apparently involves interference with the host's oxidative burst; that is, the porin allows the pathogen to evade killing by host-produced ROS (*Lorenzen et al., 2000*). Although the exact function of Endoriftia porin has not been elucidated yet, we suggest that it may have a similar function in resistance against host stress or ROS. This would be in line with elevated levels of ROS scavengers in small *Riftia* symbionts (see above). Porins are furthermore not only known to be involved in recognition by the host (e.g. in the squid symbiont *Vibrio fischeri*, *Nyholm et al., 2009*), but were also shown to be involved in survival in and communication with the host in other intracellular and pathogenic bacteria, rendering *Vibrio cholerae* and *Xenorhabdus nematophila* more resistant against antimicrobial compounds (*Mathur and Waldor, 2004*; *van der Hoeven and Forst, 2009*). As *Riftia* trophosome tissue has antimicrobial effects (*Klose et al., 2016*), and considering that *Riftia* might employ histone-derived antimicrobial peptides to modulate the symbiont's cell division (*Hinzke et al., 2019*), Endoriftia porin may enable the symbionts to reject antimicrobial compounds produced by the host. This would be of particular importance for small symbionts, as it would ensure survival of the symbiont 'stem cell' subpopulation and sustain their division capability.

Besides porin, the symbiont's outer membrane efflux pump TolC was also most abundant in fraction XS, suggesting that it may play a similar role in host interaction or persistence. TolC is a versatile export protein of Gram-negative bacteria, which interacts with different transporters of the cytoplasmic membrane to export proteins and drugs (reviewed in *Koronakis et al., 2004*). In *Sinorhizobium meliloti*, TolC is apparently involved in establishing the symbiosis with legumes, possibly by conferring increased stress resistance and by secreting symbiosis factors (*Cosme et al., 2008*), whereas *Erwinia chrysanthemi* TolC enables re-emission of the antimicrobial compound berberine and is thus essential for *Erwinia* growth in plant hosts (*Barabote et al., 2003*).

Microbe-host interactions with particular relevance in smaller Endoriftia may furthermore also be mediated by chaperones and stress proteins, which were most abundant in fraction XS (see above). Chaperones have been shown to play a role in host interaction and in intracellular survival in several pathogenic and symbiotic bacteria. For example, DnaK appears to be essential for growth of *Brucella suis* in phagocytes (*Köhler et al., 1996*), while HtpG seems to be involved in virulence and intracellular survival of *Leptospira* (*King et al., 2014*), *Salmonella* (*Verbrugghe et al., 2015*), and *Edwardsiella tarda* (*Dang et al., 2011*). Mutations in the post-transcriptional regulator *hfq* often lead to reduced fitness and virulence in bacterial pathogens (reviewed in *Chao and Vogel, 2010*). Moreover, ClpB in *Listeria* is apparently specifically involved in virulence (*Chastanet et al., 2004*), as are

ClpX and ClpP in *Staphylococcus aureus* (*Frees et al., 2003*). In the insect symbiont *Wolbachia*, HU beta was suggested to directly interact with the host (*Beckmann et al., 2013*). Additional symbiont proteins that may protect small *Endoriftia* from host interference, and particularly so in S-depleted *Riftia* specimens, included an ankyrin protein and an FK506-binding protein (see Appendix section B).

### Interaction-specific host proteins

We detected a number of *Riftia* proteins which were co-enriched with symbiont cells in fractions XS and/or S and may therefore potentially be 'symbiosis-specific'. Although we cannot entirely exclude the possibility that non-symbiont-related host organelle-bound proteins may also have accumulated in this fraction (e.g. the ribosomal and mitochondrial proteins we identified), it appears very likely that the candidates discussed below may facilitate direct host-microbe interactions or enable the host to provide optimal conditions for the symbiont. (1) Peptidoglycan-recognition proteins, for example, are involved in innate immunity (*Kang et al., 1998*) and have previously been shown or suggested to participate in symbiotic interactions (*Troll et al., 2009*; *Wang et al., 2009*; *Royet et al., 2011*; *Wippler et al., 2016*). (2) Since oxygen concentrations in the trophosome might be comparatively low (benefitting the microaerophilic symbionts; *Hinzke et al., 2019*), the hypoxia up-regulated *Riftia* proteins we detected may present a protective adaptation of the host to these hypoxic conditions. In support of this idea, Hyou1 was shown to have a protective function during hypoxia in human cells (*Ozawa et al., 1999*). (3) Moreover, enrichment of beta carbonic anhydrase 1, which interconverts bicarbonate and $CO_2$ (*Goffredi et al., 1999*; *De Cian et al., 2003*), suggests that this host protein serves to optimally provide the symbionts with $CO_2$ for fixation. (4) The host transmembrane protein 214-B (TMP214-B), which was exclusively detected in symbiont-enriched fractions (but not in trophosome homogenate) may be involved in cell death of symbiont-containing bacteriocytes by an apoptosis-related mechanism. This would be in line with our previous suggestion that apoptosis-related proteins may play a role in symbiont and bacteriocyte cell death (*Hinzke et al., 2019*) and is further supported by the fact that TMP214-B was shown to be involved in apoptosis caused by endoplasmic reticulum stress (*Li et al., 2013*). (5), Finally, the detection of degradation proteins such as cathepsin Z, legumain, glucoamylase 1, and lysosomal alpha-glucosidase in fractions XS and S may imply that the host digests not only large symbiont cells in the degradative trophosome lobule zone (see *Figure 5A*), but that small symbionts might also be exposed to host digestion (although we did not see any evidence for this in the TEM images).

## Metabolic diversity among symbiont size classes

### Large symbionts focus on carbon fixation and biosynthesis

Highest individual abundances of various carbon fixation and biosynthesis-related enzymes as well as highest overall abundances of all biosynthetic categories (including carbon-, amino acid-, lipid-, nitrogen-, and cofactor metabolism; *Supplementary file 5*) in fraction L suggest that large Endoriftia cells are relatively more engaged in the production of organic material than smaller symbiont cells. In support of this idea, we (1) observed notably higher RubisCO mRNA signal intensity in large symbiont cells than in smaller *Riftia* symbionts in our HCR-FISH analysis (Appendix section D, *Appendix 1—figure 3*). (2) Most cofactor- and vitamin metabolism-related proteins were more abundant in fractions M and/or L than in fractions XS or S. Moreover, (3) higher abundances of glycogen-producing enzymes in fraction L suggest that large symbionts invest relatively more of their biosynthetic capacities in storage of fixed carbon in the form of glycogen than smaller symbionts.

These observations concur with an autoradiographic study of *Bright et al., 2000*, who observed highest $^{14}C$ carbon incorporation in the *Riftia* trophosome lobule periphery and lowest short-term incorporation in the lobule center, and with results of *Sorgo et al., 2002*, who noted a glycogen gradient in the symbiont cells, with increasing glycogen density from the lobule center toward the periphery, that is, toward larger symbiont cells. Interestingly, in addition to our above observation that overall carbon fixation appears to play a more prominent role in larger symbionts, we also found clear indications that individual contributions of the two autotrophic pathways differ notably between symbionts of different sizes, as suggested by very dissimilar relative $\delta^{13}C$ values in our SIF analysis (Appendix section D).

A possible explanation for all these findings would be a substrate gradient caused by the direction of the blood flow from lobule periphery to lobule center (*Felbeck and Turner, 1995*; *Figure 6*, Appendix section G), which may cause carbon concentrations to be higher around the largest symbionts. While such biochemical gradients were initially suggested as the immediate cause of the symbionts' diverse morphology (*Hand, 1987*), subsequent studies instead proposed that the symbionts' polymorphism as well as metabolic differences between them result from their cell cycle (*Sorgo et al., 2002*; *Bright and Sorgo, 2003*). However, both options are not mutually exclusive, but could actually complement each other, that is, different developmental symbiont stages could face dissimilar substrate concentrations.

The putative concentration gradient, which was suggested to trigger Endoriftia's differentiation (*Bright and Sorgo, 2003*), may thus lead to differential availability of inorganic carbon (and other substrates, see below), which in turn likely results in differential regulation of bacterial gene expression, such as highest abundance of $CO_2$ incorporation enzymes in large symbionts. Large *Riftia* symbionts thus presumably not only benefit from higher $CO_2$ levels, but also have more biosynthetic capacities at their disposal than small symbionts.

This facilitates a division of responsibilities between the symbiont subpopulations: Small Endoriftia invest a considerable part of their resources in cell division and the expression of putative host interaction-related proteins that ensure survival of the 'stem cell' population (see above). In contrast, large symbionts apparently divide less frequently and may be less endangered of host interference (before they reach the degenerative lobule zone) and can thus allocate more energy to production of organic material. Such a division of labor provides the combined advantage of multiple genome copies with resulting increased transcription efficiency and productivity in large cells, and a focus on cell division in small cells. At the same time, the respective disadvantages are avoided (large Endoriftia would need to invest much more resources in cell division and replication of their multiple genome copies than small cells, while smaller Endoriftia could not produce as much biomass as large symbionts). Increased metabolic efficiency as a consequence of specialized bacteria performing complementary tasks has also been reported for different bacterial species and strains working together in other symbioses (*Zheng et al., 2019*; *Ankrah et al., 2020*). In *Riftia*, the largest symbionts that are digested at the trophosome lobule periphery are those with the highest nutritional value. They carry not only abundant carbon but also contain plenty of N-rich biomolecules in their multiple genome copies, and are likely easier to digest than smaller symbionts, due to their lower sulfur content (see below).

## Small Endoriftia store more sulfur and are more involved in sulfide oxidation

Smaller symbionts produce relatively more sulfur globules for sulfur storage than larger symbiont cells, as indicated by relatively higher abundance of sulfur globule proteins in fraction XS (*Appendix 1—figure 4*). Enclosed in a proteinaceous envelope, Endoriftia's sulfur globules contain elemental sulfur that is formed during sulfide oxidation (*Wilmot and Vetter, 1990*; *Pflugfelder et al., 2005*; *Markert et al., 2011*). Higher numbers of these globules in small symbionts are in agreement with observations of *Hand, 1987*, who noted more sulfur deposits in central (small) than in peripheral (large) *Riftia* symbionts. Although this finding was not supported by a subsequent study (*Pflugfelder et al., 2005*), our results do point to different amounts of S storage in different Endoriftia subpopulations. Storing relatively more sulfur in small Endoriftia, but less in large symbionts might aid symbiont digestion by the host (S-poor symbionts being less toxic compared to S-rich cells), a strategy that was also suggested for the thiotrophic *C. orbicularis* symbiosis (*Caro et al., 2007*). As shown for the free-living thiotrophic model bacterium *Allochromatium vinosum*, activation of stored sulfur involves trafficking proteins such as TusA, which is involved in sulfur transfer to DsrEFH and DsrC (*Stockdreher et al., 2014*). In our study, the highly abundant TusA, several DsrC copies as well as DsrEFH were all detected with highest abundances in fraction XS, thus supporting the idea of relatively more re-mobilization of sulfur and subsequent utilization of reduced sulfur compounds in small Endoriftia. As the highly abundant adenylylsulfate reductase AprAB, the ATP-sulfurylase SopT, and sulfide dehydrogenase subunit FccB were also detected with higher abundances in fractions XS or S than in M or L, one might conclude that sulfide oxidation itself also plays a more prominent role in smaller symbionts than in larger symbionts. More sulfide oxidation would also be the most straightforward explanation for the observed higher amounts of stored sulfur in small

symbionts (see above). However, as we detected the dissimilatory sulfite reductase DsrAB, the third key enzyme of cytoplasmic sulfide oxidation, with a rather ambiguous abundance pattern (*Supplementary file 3e*), this idea remains speculative and requires further analysis.

## In large symbionts, thiosulfate oxidation plays a more prominent role

Larger symbionts may rely relatively more on thiosulfate oxidation – in addition to sulfide oxidation – than smaller Endoriftia, as suggested by highest abundance of SoxZ and detection of several other (low-abundant) Sox proteins in fraction L. Although Endoriftia prefers sulfide (*Wilmot and Vetter, 1990*), it was also shown to oxidize thiosulfate, and may supplement sulfide oxidation with thiosulfate oxidation for energy production (*Robidart et al., 2011*). Since expression of the thiosulfate-oxidizing Sox (sulfur oxidation) complex was shown to be upregulated in the presence of thiosulfate in *A. vinosum* (*Grimm et al., 2011*), we speculate that higher Sox abundance in large symbionts indicates higher thiosulfate concentrations in the trophosome lobule periphery than in the lobule center. This could be due to a concentration gradient (as proposed above for $CO_2$) and/or possibly also a result of host thiosulfate production. The *Riftia* host appears to be able to oxidize toxic sulfide to the less toxic thiosulfate in its mitochondria (*Hinzke et al., 2019*). Higher abundance of host thiosulfate sulfurtransferase in symbiont-enriched fractions compared to non-enriched trophosome homogenate in our present study suggests that this putative detoxification process could be particularly important in the symbiont-containing bacteriocytes. With sulfide supposedly reaching the trophosome lobule periphery first with the blood flow, free sulfide concentrations might be higher there and, consequently, host sulfide oxidation to thiosulfate might be more frequent in bacteriocytes at the lobule periphery than in the center. The idea of more thiosulfate oxidation in large Endoriftia is further substantiated by highest abundance of six rhodanese family proteins in fraction L, as rhodanese-like proteins can cleave thiosulfate into sulfite and sulfide and were proposed to be involved in thiosulfate oxidation (*Hensen et al., 2006*; *Welte et al., 2009*). The combined use of both sulfide and thiosulfate has also been reported for chemoautotrophic symbionts of snails and mussels (*Beinart et al., 2015*). In these symbioses, thiosulfate oxidation supported carbon fixation just as effectively as sulfide (or even more effectively). Using thiosulfate as their (additional) energy source may therefore not be an energetic disadvantage for *Riftia* symbionts. Moreover, thiosulfate oxidation in Endoriftia may have the positive side-effect of reducing competition for sulfide between larger and smaller symbionts.

Interestingly, overall abundance of all proteins involved in the symbiont's energy-generating sulfur metabolism, the most abundant of all metabolic categories, remained relatively unchanged across the four fractions (*Supplementary file 5*). This indicates that sulfur oxidation-based energy generation, a fundamental basis of all other metabolic processes, is equally important throughout the symbiont's differentiation process, even if individual contributions of reduced sulfur compounds may differ. (For a detailed overview of sulfur oxidation reactions in Endoriftia see Appendix section F).

## Hydrogen oxidation is more relevant in large symbionts

In large symbionts, the use of hydrogen may furthermore play a more prominent role than in smaller symbiont cells, as suggested by increasing abundances of the Isp-type respiratory $H_2$-uptake [NiFe] hydrogenase large subunit HyaB, a Fe-S oxidoreductase (GlpC) encoded next to HyaB, and the hydrogenase expression/formation protein HypE from fraction XS to L. The small hydrogenase subunit HyaA (Sym_EGV51837.1) and an additional hydrogenase expression/formation protein (HoxM, Sym_EGV51835.1), both of which are encoded upstream of HyaB in the symbiont genome, were detected with increasing abundance toward fraction L as well (although at very low concentrations; *Supplementary file 2b*), supporting the idea of relatively more hydrogen oxidation in large symbionts. Like for $CO_2$ and thiosulfate, this might be due to a concentration gradient with highest hydrogen concentrations at the lobule periphery and lowest concentrations toward the lobule center. Use of hydrogen as an energy source has been described or suggested for free-living sulfur oxidizing bacteria like *A. vinosum* (*Weissgerber et al., 2011*), and for a variety of thiotrophic symbionts of marine invertebrates, including *Riftia* (*Petersen et al., 2011*). However, above-seafloor hydrogen concentrations in diffuse-flow fluids at our particular sampling site were reported to be very low (<2 µM; *McNichol et al., 2018*), and it is presently unclear whether and how ambient hydrogen

would reach the intracellular symbionts in the trophosome. Nevertheless, taking advantage of hydrogen oxidation in addition to sulfide- and thiosulfate oxidation, that is, using a broader repertoire of electron donors, would potentially enhance the metabolic flexibility, particularly of large Endoriftia. However, $H_2$ was recently suggested to be involved in maintaining intracellular redox homeostasis rather than working as electron donor in the *Riftia* symbiosis (*Mitchell et al., 2019*), and hydrogenase may in fact also play a role in sulfur metabolism (as suggested for *A. vinosum* (*Weissgerber et al., 2014*); Appendix section F). Therefore, the exact role of hydrogen oxidation in Endoriftia and why it might be relatively more relevant in larger symbionts remains to be discussed.

## Small and large symbionts favor separate parts of the denitrification pathway

Our results suggest that small *Riftia* symbionts rely relatively more on the NarGHI-mediated first step of anaerobic respiratory nitrate reduction to nitrite. In contrast, Nap-mediated nitrate reduction and all subsequent denitrification steps via NO and $N_2O$ to $N_2$ seem to be more prominent in larger symbionts. While Endoriftia has the genomic potential for complete denitrification to $N_2$ and was experimentally shown to use nitrate for respiration (*Hentschel and Felbeck, 1993*), our results thus point to differential use of separate parts of this pathway in small and large symbionts. Since expression of membrane-bound nitrate reductase Nar is inhibited by oxygen (*Moreno-Vivián et al., 1999*), highest NarGHI abundance in fraction XS suggests that $O_2$ levels might be particularly low around smaller symbionts. This is presumably due to an oxygen gradient from lobule periphery to center (see also Appendix section G). Following this idea, increasing abundances of NapCHG, subunits of the periplasmic nitrate reductase Nap complex, from fraction XS to L indicate that Nap could take over the role of Nar in larger symbionts. As the inhibitory effect of oxygen on denitrification was proposed to be largely due to inhibition of nitrate transport across the cytoplasmic membrane to Nar (*Denis et al., 1990*; *Moir and Wood, 2001*), the periplasmic Nap nitrate reductase is not affected by elevated oxygen concentrations (*Moreno-Vivián et al., 1999*). Some denitrifying bacteria therefore couple Nap to nitrite reductase Nir, nitric oxide reductase Nor and nitrous oxide reductase Nos under oxic or microaerophilic conditions for aerobic denitrification (*Ji et al., 2015*). We propose that large Endoriftia adopt this strategy of aerobic denitrification when exposed to higher oxygen levels at the lobule periphery. Presumed higher nitrate concentrations around large Endoriftia may additionally facilitate increased expression of Nap, Nir, Nor, and Nos (compared to smaller symbionts). Moreover, while Nar is an energy-conserving enzyme and generates a proton motive force, which is potentially advantageous during cell division in small symbionts (see Appendix section B), the periplasmic Nap conserves less energy and does not contribute to the proton gradient (*Bell et al., 1993*). Only larger symbionts may therefore be able to afford Nap.

While the above scenario of anaerobic nitrate reduction in small Endoriftia and aerobic denitrification in large Endoriftia seems quite plausible in the context of our findings, some aspects of this hypothesis are still unclear. Nar, Nap, and Nir may not be exclusively involved in respiration, but could as well operate in nitrate assimilation (*Liao et al., 2014*; *Hinzke et al., 2019*). Highest NarGHI and Nap abundances in small and large Endoriftia, respectively, may therefore be due to either of the two functions or (more likely) a combination of both. Also, the fate of nitrite produced by NarGHI during dissimilatory nitrate reduction in small symbionts is dubious. Nitrite is toxic and needs to be removed from the cells either by conversion into ammonium or by export (*Moir and Wood, 2001*). Although assimilation appears to be the most likely option, Endoriftia's assimilatory nitrite reductase was not detected in fractions XS and S (but only in fractions M and L and at extremely low abundance; large subunit NirB) or not detected at all (small subunit NirD, Sym_2601635198). Similarly, the nitrite extrusion protein NarK had its lowest abundance in fraction XS. Whether nitrite assimilation in Endoriftia relies on enzymes other than NirBD, or whether nitrite is removed from the small symbionts by other exporters or by diffusion, and possibly even passed on to large Endoriftia for further reduction, therefore needs to be elucidated in future studies.

## Regulation of gene expression may be less stringent in large symbionts

Relative abundance of the RNA polymerase sigma factor RpoD decreased from fraction XS to L (*Figure 3*, *Supplementary file 3h*) in S-rich and in S-depleted samples, pointing to relatively more growth-related activities in small Endoriftia (see also Appendix section B). RpoD is the primary sigma

factor for vegetative growth (σ70), which regulates transcription of most genes involved in exponential growth in many bacteria (*Helmann and Chamberlin, 1988*; *Fujita et al., 1994*; *Ishihama, 2000*). This would be in agreement with the idea of small *Riftia* symbionts being mainly occupied with cell division and proliferation in a quasi-exponential growth phase, while large symbionts function as biosynthetic 'factories', focusing on carbon fixation and biomass production. Interestingly, RpoS, the master transcriptional regulator of stationary phase gene expression and antagonist of RpoD, was not detected in any of our samples (although it is encoded in the symbiont genome). RpoS abundance increases upon stress and limitation during transition to the stationary phase in free-living model bacteria (*Hengge-Aronis, 1993*; *Fujita et al., 1994*; *Ishihama, 2000*). Its absence in the *Riftia* symbiont's proteome suggests that, unlike free-living bacteria, the symbiont does not experience a stationary phase-like growth arrest even in later developmental stages, probably because it is ideally supplied with all necessary substrates by the host. The symbiont likely adapts to this 'lack' of stress or limitation by less stringent regulation of gene expression, which could explain the metabolic diversity we observed particularly in large symbionts, such as multiple ways of energy generation (thiosulfate- and hydrogen oxidation in addition to sulfide oxidation) and two $CO_2$ fixation pathways. Calvin cycle and rTCA cycle are in fact in all likelihood expressed in the very same (large) symbiont cells, as suggested by our HCR-FISH experiments (*Appendix 1—figure 3*). Under these premises, the previously observed simultaneous expression of such seemingly redundant metabolic pathways in *Riftia* symbionts (*Markert et al., 2011*) very likely reflects this presumptive 'de-regulation' of gene expression in large parts of the symbiont population (i.e. in the large symbionts), which allows Endoriftia to fully exploit its versatile metabolic repertoire to the advantage of the symbiosis.

## Conclusion

Our results show that Endoriftia cells of different differentiation stages likely employ distinct metabolic profiles, thus confirming our initial hypothesis. The symbionts divide responsibilities: Whereas small Endoriftia ensure survival of the symbiont population, large Endoriftia are primarily engaged in biomass production.

The factors that trigger Endoriftia's development from smaller to larger cells remain to be elucidated. For *Rhizobium*, a steep $O_2$ concentration gradient inside legume nodules was proposed to be involved in signaling for symbiont differential gene expression (*Soupène et al., 1995*). Similarly, some of the differences we observed in small and large Endoriftia might also be connected to the availability of electron donors or acceptors, and hence differentiation of Endoriftia cells might depend on substrate availability. Symbiont differentiation in *Riftia* might furthermore be induced by specific host effectors, for example, histone-derived antimicrobial peptides, which were recently proposed to play a role in symbiont cell cycle regulation (*Hinzke et al., 2019*), or other compounds that allow *Riftia* to modulate the symbiont's expression of certain metabolic pathways. Besides such direct interference, *Riftia* likely also exerts indirect influence on symbiont gene expression by providing copious amounts of all necessary substrates to the bacterial partner.

We speculate that this constantly high nutrient availability inside the host causes Endoriftia's biosynthetic pathways to be regulated less stringently (compared to what we would expect in free-living bacteria). This would explain the previously observed metabolic versatility of symbionts in the same host: Large Endoriftia can afford to employ multiple – even redundant – metabolic pathways at the same time. Division of labor between subpopulations, which enables such an 'advantageous deregulation' in specialized cells, thus likely enhances symbiont productivity during symbiosis. We suggest that such symbiont differentiation into functionally dissimilar subpopulations with complementary roles in the same host might be an important key to success of microbe-host associations in various environments.

## Materials and methods

### Sample collection and enrichment of symbiont subpopulations

*Riftia* samples for enrichment of symbiont subpopulations were collected at the East Pacific Rise hydrothermal vent field at 9°50′ N, 104°17′ W in a water depth of about 2500 m during a research cruise with R/V Atlantis in November 2014 (AT26-23). Samples for electron microscopy were obtained during a second cruise (AT37-12) at the same site during March-April 2017 (*Hinzke et al.,*

*2019*). Sample details and numbers of biological replicates are summarized in *Supplementary file 1*.

Trophosome sulfur content of the specimens was estimated based on the trophosome tissue's color: sulfur-rich (S-rich) specimens have a light green or yellowish trophosome, due to the sulfur stored in the symbionts, whereas trophosomes of sulfur-depleted (S-depleted) specimens appear dark green to black (*Pflugfelder et al., 2005*). For proteomic analyses, we used seven healthy-looking *Riftia* specimens, of which four had S-rich trophosomes and three had S-depleted trophosomes. (Note that sulfur content does not provide any information on the symbiosis' state of health, that is, worms with S-depleted trophosomes are not 'unhealthier' than worms with S-rich trophosomes.) Since trophosome sulfur content is positively correlated with habitat sulfide concentrations (*Childress et al., 1991*; *Robidart et al., 2011*; *Scott et al., 2012*), it can serve as an indicator for the energetic situation of the symbiosis at the time of sampling.

To enrich symbiont cells of varying sizes (i.e. morphologically distinct symbiont subpopulations), *Riftia* specimens were dissected onboard the research vessel immediately after recovery of the worms. Approximately 3 ml trophosome tissue were homogenized in a Dounce glass homogenizer in 6 ml imidazole-buffered saline (IBS, 0.49 M NaCl, 0.03 M MgSO$_4$, 0.011 M CaCl$_2$, 0.003 M KCl, 0.05 M imidazole). As described in *Hinzke et al., 2018*, the homogenate was subjected to rate-zonal density gradient centrifugation, which allows to separate particles based on their size (*Graham, 2001*). In brief, an 8–18% Histodenz density gradient was created using a dilution series of Histodenz in IBS (1% steps, 1 ml per step), which was stacked in a 15 ml tube so that Histodenz concentration was highest at the bottom and lowest at the top. On top of this gradient, 0.5 ml tissue homogenate was layered and the gradient was centrifuged (1000 x g, 5 min, 4°C) in a swing-out rotor. Smaller symbiont cells were thus enriched in less dense gradient fractions (lower Histodenz concentrations) in the upper part of the gradient, while larger cells migrated to lower gradient fractions. After centrifugation, gradients were disassembled by carefully fractionizing the entire gradient volume into 0.5 ml subsamples, giving a total of 24 fractions. Enrichment of distinct symbiont subpopulations in these subsamples was confirmed using catalyzed reporter deposition-fluorescence in situ hybridization (CARD-FISH, see below). For this purpose, 20 µl of each gradient fraction subsample and 15 µl of homogenate was fixed in 1% PFA in IBS, and symbiont cells were subsequently filtered onto GTTP polycarbonate filters (pore size 0.2 µm, Millipore) as described previously (*Ponnudurai et al., 2017*).

## CARD-FISH

Enrichment of symbiont cell sizes in gradient fractions was analyzed employing fluorescence microscopy with samples labeled by CARD-FISH in five biological replicates (*Riftia* specimens) with S-rich trophosome and three biological replicates with S-depleted trophosomes. CARD-FISH labeling was performed as previously described (*Ponnudurai et al., 2017*), using the probe Rif445 (*Nussbaumer et al., 2006*) and Alexa Fluor 594-labeled tyramide. For counterstaining, 0.1% (w/v) 4,6-diamidino-2-phenylindole (DAPI) was added to the embedding medium (4:1 Citifluor AF1 (Citifluor) and Vectashield (Vector Laboratories)). CARD-FISH filters were analyzed using an Axio Imager.M2 fluorescence microscope (Carl Zeiss Microscopy GmbH). For semi-automated cell counting and to measure the longest cell dimension, we used a custom Fiji (*Schindelin et al., 2012*) macro with the Fiji plugins Enhanced Local Contrast (CLAHE; *Saalfeld, 2010*) and Bi-exponential edge preserving smoother (BEEPS; *Thévenaz et al., 2012*). After image processing, we excluded objects with a size of less than 2 µm (as these were mainly artifacts) and set the maximum object size to 20 µm. To assign cell sizes to size classes (i.e. cell size ranges), we used a quartile split: We calculated quartiles of cell sizes in non-enriched homogenate samples (i.e. 25% of all cells in homogenate samples were assigned to each class). This resulted in the four calculative size classes very small (≥2 µm –<3.912 µm), small (≥3.912 µm –<5.314 µm), medium (≥5.314 µm –<6.83275 µm), and large (≥6.83275 µm – 20 µm). The majority of cells in all size classes were coccoid. Rod-shaped cells were almost exclusively present in the smallest size class. Individual gradient fractions (subsamples) were screened for their respective share of cells in each size class and the subsample with the highest percentage of cells in the respective quartile was chosen for metaproteomic analysis. For example, if of all 24 subsamples of a sample, the fifth fraction (counted from the top of the gradient) had the highest percentage of very small cells, that is, most of the cells in fraction 5 were between 2 µm and 3.912 µm in diameter (as measured by our Fiji macro), this fraction was chosen as representative of

very small symbiont cells in the respective biological replicate (worm). The fraction containing the highest percentage of very small cells is referred to as 'fraction XS' in the manuscript. The fractions containing the highest percentages of small, medium, and large symbiont cells are referred to as 'S', 'M', and 'L', respectively. If the same subsample had the highest percentage of cells in two size classes, this subsample was chosen as representative for one of these size classes, and for the other size class, the subsample with the second highest percentage of cells in that class was used as representative. Cell size distributions in the four size class representatives are summarized in *Figure 1*. For each biological replicate, fractions XS, S, M, and L, as well as non-enriched trophosome homogenate were subjected to separate metaproteome analyses (see below).

## Transmission electron microscopy (TEM)

Trophosome samples used for TEM in this study (see *Supplementary file 1* for details) were prepared and analyzed as described previously (*Hinzke et al., 2019*). Tissue sections were recorded on sheet films (Kodak electron image film SO-163, Plano GmbH, Wetzlar) as described by *Petersen et al., 2020*. To create a composite high-resolution TEM image of a trophosome lobule (*Figure 5A*), we merged 50 individual micrographs of one section using Serif Affinity Photo (https:// affinity.serif.com/en-us/photo/). All 50 partially overlapping images were loaded and the fully automated 'Panorama Stitching' technique was applied, resulting in a panorama image still showing some vignette marks caused by inhomogeneous exposure at the former edges of individual images. The global smooth frequencies reflecting these exposure errors were removed using the frequency separation filter with a large radius. The gradation curve was manually corrected. For acquisition of the images in *Figure 5B*, a wide-angle dual speed CCD camera Sharpeye (Tröndle, Moorenweis, Germany) was used, operated by the ImageSP software. All micrographs were edited using Adobe Photoshop CS6.

## HCR-FISH and confocal laser scanning microscopy (CLSM)

A gradient fraction enriched in large symbiont cells (see *Supplementary file 1* for details) that was fixed for CARD-FISH and immobilized on GTTP polycarbonate filters as described above was used for hybridization chain reaction FISH (HCR-FISH) according to *Choi et al., 2014*. We used an HCR-FISH v2.0 Custom Kit (Molecular Technologies) according to the manufacturer's instructions. Probes targeted the *Riftia* symbiont's 16S rRNA (fluorescence marker: Alexa Fluor 488), and the mRNAs of ATP-citrate lyase subunit AclB (Alexa Fluor 647) and RubisCO (Alexa Fluor 594). Assisted by the probe manufacturer, we designed five individual probes per target RNA (for 16S and AclB) and four probes for RubisCO, each of which targeted a 50 nucleotides long sub-sequence of the respective RNA of interest. The sub-sequences were spread out across the length of each target RNA (see *Appendix 1—table 1* for the probe sequences). In brief, filter sections were washed twice with 50% hybridization buffer (50% formamide, 5x sodium chloride sodium citrate (SCSC, 0.75 M NaCl, 75 mM $Na_3C_6H_5O_7$), 9 mM citric acid, pH 6.0, 0.1% Tween 20, 50 µg/ml heparin, 1x Denhardt's solution, 10% dextran sulfate) in 2x mPBS (89.8 mM $Na_2HPO_4$, 10.2 mM $NaH_2PO_4$, 0.9 M NaCl) at 45°C for 30 min for pre-hybridization, and incubated overnight (16 hr, 45°C) with probe solution (1 pmol of each probe in 500 µl hybridization buffer). Excess probes were removed with several washing steps in 75–25% probe wash buffer (50% formamide, 5x SCSC, 9 mM citric acid, pH 6.0, 0.1% Tween 20, 50 µg/ml heparin in 5x SCSC) for 15 min at 45°C, 300 rpm, and subsequently in 5x SCSC for 30 min at 45°C and 300 rpm. Samples were pre-amplified with DNA amplification buffer (5x SCSC, 0.1% Tween 20, 10% dextran sulfate). Hairpins were activated by snap-cooling and added to the samples. After overnight incubation (16 hr, room temperature) with the hairpin solution, samples were washed with 5x SCSC, containing 0.05% Tween 20 (room temperature, 300 rpm, four times 5 min, two times 30 min), and embedded in Mowiol 4–88 (Carl Roth GmbH) embedding medium prepared according to the manufacturer's instructions. Confocal microscopy was performed on a Zeiss LSM510 meta equipped with a 100x/1.3 oil immersion objective. Probes were excited with laser lines 633 (ATP-citrate lyase), 561 (RubisCO) and 488 (16S rRNA) and signals were detected with filters suitable for dye maximal emissions at 670 nm, 595 nm and 527.5 nm, respectively. Signal intensities and cell sizes (from eight frames showing a total of 33 cells on a filter from one biological replicate, n = 1) were quantified using the Fiji software package (*Schindelin et al., 2012*). Individual cells were defined as regions of interest (ROI), in which signal intensity per pixel was recorded. Mean pixel intensity of ROI

was calculated and background was corrected. Global background values were calculated for every channel based on up to six ROIs randomly placed in each image frame. The following cell size parameters were calculated: (i) Feret's diameter (the longest distance between any two points along the boundary of the ROI) and (ii) the area of the ROI. As described by *Nikolakakis et al., 2015*, two negative control preparations were performed to identify artifact signals: Two filters from the same biological replicate as above were incubated (a) without probes but with fluorophore-carrying hairpins (n = 1; to test for non-specific amplification of hairpins) and (b) without probes or hairpins (n = 1; to test for autofluorescence), and analyzed using an Axio Imager.M2 fluorescence microscope (Carl Zeiss Microscopy GmbH). Since Endoriftia is uncultivable as yet, a target-free control sample (i.e. an Endoriftia knockout mutant without the respective RNAs of interest) is not available at this point. Nonspecific binding of the probes to non-target RNAs was therefore excluded based on the following considerations: (i) In nucleic acid hybridizations, the hybridization rate decreases to zero when mismatches increase to more than 30% (*Wetmur, 1991*). (ii) There is only one symbiont phylotype in *Riftia*, preventing false-positive binding to other bacterial RNAs, and (iii) false-positive matches to host RNAs could be excluded, since symbiont cells were co-labeled with the 16S rRNA-specific probe which confirmed the identity of the cells. Finally (iv), to ensure probe sequence specificity, we blast-searched all probe candidates against the *Riftia* symbiont genome (NCBI whole-genome shotgun sequences NZ_AFOC00000000.1 and NZ_AFZB00000000.1) at low stringency (BLASTn optimized for somewhat similar sequences, seed length: 7, expectation threshold: 1), and subjected all resulting matches to in silico hybridization behavior analyses using the mathFISH web tool (*Yilmaz et al., 2011*). These calculations confirmed that in *Riftia* symbiont cells, either of the two mRNA probes has only one target sequence to which it binds with a hybridization efficiency of 100%, whereas hybridization efficiency of off-target sequences was 0% (data not shown).

## Flow cytometry

Subsamples of fresh homogenate, of three gradient fractions enriched in small symbionts, and of three fractions enriched in large symbionts were fixed in 1% PFA as for CARD-FISH (see above) in two biological replicates (i.e. from two *Riftia* specimens). Right before flow cytometry analysis, fixed cells were carefully pelleted and incubated in 0.1 mg/ml RNAse A (from bovine pancreas, DNase-free, Carl Roth, Germany) for 30 min at 37°C to remove RNA, and stained with Syto9 (final concentration 0.5 μmol/l in PBS), a dye that selectively stains DNA and RNA (*Stocks, 2004*). The fluorescence signal was analyzed using a FACSAria high-speed cell sorter (Becton Dickinson Biosciences, San Jose, CA, USA) with 488 nm excitation from a blue Coherent Sapphire solid state laser at 18 mW. Optical filters were set up to detect the emitted Syto9 fluorescence signal at 530/30 nm (FITC channel). All fluorescence data were recorded at logarithmic scale with the FACSDiva 8.02 software (Becton Dickinson). Prior to measurement of experimental samples, the proper function of the instrument was determined by using the cytometer setup and tracking software module (CS and T) together with the CS and T beads (Becton Dickinson Biosciences). During sample measurements, the present populations were shown in a side scatter (SSC)-area versus forward scatter (FSC)-area dot plot. The detection thresholds and photomultiplier (PMT) voltages were adjusted by using an unstained sample. The Syto9 signal from the scatter populations was monitored in a Syto9-area histogram. For each sample at least 10,000 events in the scatter gate were recorded. For further analysis, the Syto9-stained bacteria (populations 1 and 2, see *Figure 2*) were sorted from the bivariate dot plot, SSC versus Syto9 (FITC-channel). Prior to sorting, the proper function of the cell sorter was determined using the AccuDrop routine. Data analysis was done with the software FlowJo V10. To evaluate the results of the sorting procedure, FACS-sorted cell populations as well as unsorted subsamples of homogenate and gradient fractions were examined using an Axio Imager.M2 fluorescence microscope (Carl Zeiss Microscopy GmbH).

## Peptide sample preparation

For four biological replicates with S-rich trophosome and three biological replicates with S-depleted trophosome, non-enriched trophosome homogenate and the four density gradient fractions determined as XS, S, M, and L by CARD-FISH analyses (see above) were individually subjected to metaproteomic analyses. Proteins were extracted as described in *Hinzke and Markert, 2017*. Briefly, cells were mixed with lysis buffer (1% (w/v) sodium deoxycholate (SDC), 4% (w/v) sodium dodecyl

sulfate (SDS) in 50 mM triethylammonium bicarbonate buffer (TEAB)), heated for 5 min at 95°C and 600 rpm and cooled on ice. Samples were then placed in an ultrasonic bath for 5 min and subsequently cooled on ice. Cell debris was removed by centrifugation (14,000 x g, 10 min, room temperature). Protein concentration was determined using the Pierce BCA assay according to the manufacturer's instructions. Peptides were generated using a 1D gel-based approach as in *Ponnudurai et al., 2017* with minor modifications. In brief, 20 µg of protein sample was mixed with Laemmli sample buffer containing DTT (final concentration 2% (w/v) SDS, 10% glycerol, 12.5 mM DTT, 0.001% (w/v) bromophenol blue in 0.06 M Tris-HCl; *Laemmli, 1970*) and separated using precast 4–20% polyacrylamide gels (BioRad). After staining, protein lanes were cut into 10 pieces, destained (600 rpm, 37°C, 200 mM $NH_4HCO_3$ in 30% acetonitrile), and digested with trypsin (sequencing grade, Promega) overnight at 37°C, before peptides were eluted in an ultrasonic bath. Peptides were stored at -80°C until LC-MS analysis.

## LC-MS/MS analysis

MS/MS measurements were performed as described previously by *Ponnudurai et al., 2017*. In brief, samples were measured with an LTQ-Orbitrap Velos mass spectrometer (Thermo Fisher, Waltham, MA, USA), coupled to an EASY-nLC II (ThermoFisher) for peptide separation using a 100 min binary gradient. MS data were acquired in data-dependent MS/MS mode for the 20 most abundant precursor ions. After a full scan in the Orbitrap analyzer (R = 30,000), ions were fragmented via CID and recorded in the LTQ analyzer. Samples were measured in a randomized design.

## Protein identification and function prediction

Proteins were identified by searching the MS/MS spectra against the *Riftia* host and symbiont database (*Hinzke et al., 2019*), which was constructed from the host transcriptome and three symbiont genome assemblies, that is, NCBI project PRJNA60889 (endosymbiont of *Riftia pachyptila* (vent Ph05)), NCBI project PRJNA60887 (endosymbiont of *Tevnia jerichonana* (vent Tica)), and JGI IMG Gold Project Gp0016331 (endosymbiont of *Riftia pachyptila* (vent Mk28)). The cRAP database containing common laboratory contaminants (*The Global Proteome Machine Organization, 2017*) was added to complete the database. Database search was conducted using Proteome Discoverer v. 2.0.0.802 as described in *Kleiner et al., 2018*. Briefly, raw spectra were searched against the database using the Sequest HT node. False discovery rates (FDRs) for peptide spectrum matches were calculated and filtered using the Percolator Node (FDR < 0.05). FidoCT was used to infer proteins with a protein-level false discovery rate of 5% (q-value <0.05, at least one unique peptide). The mass spectrometry proteomics data have been deposited to the PRIDE proteomics identification database (https://www.ebi.ac.uk/pride/; *Vizcaíno et al., 2016*) with the dataset identifier PXD016986.

To systematically screen the *Riftia* symbiont metagenome for dissimilatory sulfur metabolism-related proteins, candidates identified in different studies were searched against the Endoriftia metaproteome database using bioedit (*Hall, 1999*; *Supplementary file 7*). Host proteins were additionally annotated using the same tools as in *Hinzke et al., 2019*. Symbiont hydrogenase sequences were classified using HydDB (*Søndergaard et al., 2016*).

Relative stable isotope fingerprints (relative SIFs, relative $\delta^{13}C$ values) of all S-rich (n = 4) and S-depleted (n = 3) gradient fractions were calculated using Calis-p 2.0 (*Kleiner et al., 2018*), with fraction XS of S-rich trophosome as baseline.

## Statistical evaluation of metaproteomics data and abundance quantification

### Filtering and normalization

For samples from sulfur-rich specimens, four replicates for each of the four size classes were used (resulting in 16 samples); for analysis of symbionts from sulfur-depleted specimens, three replicates were available per size class (giving a total of 12 samples). For comparisons of protein abundance (i) across different samples, for example, to determine a protein's abundance trend across gradient fractions XS to L, edgeR-RLE-normalized spectral counts were calculated (see below), while (ii) %orgNSAF values were used for abundance comparisons of different proteins within one sample, for example, to determine the 'most abundant' proteins in a sample.

i. To allow for comparisons of protein abundance across different samples, spectral count data were first filtered so that they included only proteins that had at least five spectral counts in at least four out of 16 (S-rich specimens) or three out of 12 (S-depleted specimens) samples. The filtered dataset was then normalized using Relative Log Expression (RLE) normalization with the package edgeR v.3.24.3 (*Robinson et al., 2010*) in R v. 3.5.1 (*R Development Core Team, 2018*; *Supplementary file 2a*). The filtering and normalization step was included to avoid biasing the analysis toward symbiont proteins that were only detected in the high-density fractions M and L (enriched in larger symbiont cells), but which were absent in fractions of lower density (XS and S, containing primarily smaller cells). Fractions S and particularly XS contained relatively more host proteins, leading to a lower total number of detectable symbiont proteins. (Note that these values were not normalized to protein size, so that a protein's relative abundance changes can be followed across different samples, but abundances cannot be compared between proteins). We tested for significant differences in symbiont protein abundance between individual gradient factions (representing enrichments of different cell size classes) using two methods, that is, profile clustering (STEM analysis) and random forests (see below).

ii. To be able to compare relative symbiont protein abundances within samples and to identify particularly abundant proteins, normalized spectral abundance factor (NSAF) values were calculated from unfiltered spectral counts by normalization to protein size and to the sum of all proteins in a sample (*Zybailov et al., 2006*; *Mueller et al., 2010*). %orgNSAF values give an individual protein's percentage of all proteins of a given organism ('org') in the same sample (*Supplementary file 2b*). Note that %orgNSAF values in this analysis cannot be compared across different samples, due to the unequal number of total host and symbiont proteins in different samples.

## STEM analysis

For protein expression profile clustering, we employed the Short Time Series Expression Miner (STEM; *Ernst and Bar-Joseph, 2006*) v. 1.3.11., which fits gene expression profiles in ordered short series datasets (like the cell cycle stages of *Ca.* E. persephone), to model profiles representing different expression patterns. Filtered and RLE-normalized data were log-normalized, repeat data were defined to be from different time points and data were clustered using the STEM method with default options. For STEM filtering, the minimum correlation between repeats and the minimum absolute expression change were set to 0.5. All permutations were used. For correction, the false discovery rate (FDR) was set to 0.05. Profiles were clustered with a minimum correlation percentile of 0.5. Other parameters were left at default values. Proteins which were assigned to model profiles, that is, all proteins which were not removed by filtering and showed a consistent trend in all replicates, were used for further analysis. This means that differences in protein abundance patterns were considered significant if proteins were detected with a consistent abundance trend across all replicates (increase, decrease, or alternating increase and decrease of abundance from fraction XS to L).

## Random forests

For random forest analysis, we used the ranger package v. 0.10.1 (*Wright and Ziegler, 2015*) in R v. 3.5.1 (*R Development Core Team, 2018*). Random forests are a machine learning technique, which can be used to find the variables – here proteins – that allow to predict which datasets or samples are similar (and which ones are not; *Degenhardt et al., 2019*). For variable importance calculation, we employed the method from *Janitza et al., 2018* as implemented in the ranger package. This method uses a heuristic approach, where a null distribution for p-value calculation is generated based on variables with importance scores of zero or negative importance scores. For pairwise comparisons, the data set was subjected to an additional filtering step, so that only proteins with a minimum of five spectral counts in at least six out of eight (S-rich) or four out of six (S-depleted) samples were included. The comparison of all 16 S-rich samples included only such proteins which had a minimum of five spectral counts in at least five samples, and the comparison of all twelve S-depleted samples included only proteins with five or more spectral counts in a minimum of four samples. The filtered and RLE-normalized data were used for random forest analysis as follows: 2000 forests with 10,000 trees per forest were grown for pairwise comparisons as well as for comparisons including

the samples representing all four size classes. Proteins which had a p-value below 0.05 in >90% of the forests were included in further analyses.

## Significant differences in protein abundance

Proteins that showed significant abundance differences as determined by STEM analysis or by random forest analysis (see above) or by both methods were included in a common list. Please note that this approach of determining significant protein abundance differences was not based on individual p-values. For proteins with significant abundance differences, we clustered the z-scored mean abundances using hierarchical clustering (Pearson correlation, complete linkage) in R to visualize their abundance trends (*Appendix 1—figure 1*). For this purpose, we employed the R base package stats (*R Development Core Team, 2018*) as well as the packages cluster (*Maechler et al., 2018*) and ComplexHeatmap (*Gu et al., 2016*). For comparison of S-rich and S-depleted symbionts of the same size class, we used the R package edgeR v. 3.24.3 (*Robinson et al., 2010*), which uses a Bayes-moderated Poisson model for count data analysis, with an overdispersion-adapted analogon to Fisher's exact test for detecting differentially expressed genes (*Robinson et al., 2010*).

## Host proteins

Host proteins which were more abundant in symbiont-enriched fractions as compared to the non-enriched trophosome homogenate are candidates for direct host-symbiont interaction, as they might be secreted into symbiont compartments or even physically associated with symbiont cells. For evaluation of host protein enrichment, we used fractions XS and S, enriched in the two smallest symbiont size classes (i.e. fractions collected from the upper part of the gradient). As the lower gradient fractions sometimes contained the gradient pellet, in which host proteins can also accumulate when host tissue fragments are pelleted, these fractions were not used for host protein analysis. Comparisons of relative host protein abundance between trophosome homogenate and fractions XS and S were performed using the R package edgeR v. 3.24.3. Spectral count data were filtered to include only proteins which had at least five spectral counts in at least four (for S-rich specimens) or three (in S-depleted specimens) samples and RLE-normalized abundance values were compared between samples. Proteins which were significant in the edgeR comparison and had a higher mean RLE-normalized abundance in fractions XS and S than in the homogenate sample were included in further analysis.

## Acknowledgements

We thank captains and crews of R/V Atlantis and DSV Alvin who supported sampling during cruises AT26-23 and AT37-12. We are grateful to Jana Matulla and Annette Meuche for excellent technical assistance in sample preparation for proteomics and electron microscopy, respectively. Thanks to Ruby Ponnudurai and Frank Unfried for help with CARD-FISH, and to Alexander Graf, Mathis Appelbaum, Judith Zimmermann and Silke Wetzel for advice on epifluorescence microscopy and staining. We greatly appreciate Elisa Kasbohm's help with random forest analyses. We are very grateful to Jörg Bernhardt for stitching the transmission electron micrographs to produce a panorama image with high resolution. This work was supported by the German Research Foundation DFG (grant MA 6346/2–1 to SM), fellowships of the Institute of Marine Biotechnology Greifswald (TH, MM), a German Academic Exchange Service (DAAD) grant (TH), the NC State Chancellor's Faculty Excellence Program Cluster on Microbiomes and Complex Microbial Communities (MK), the USDA National Institute of Food and Agriculture, Hatch project 1014212 (MK), the U.S. National Science Foundation (grants OCE-1131095 and OCE-1559198 to SMS), and The WHOI Investment in Science Fund (to SMS). We furthermore acknowledge support for article processing charges from the DFG (Grant 393148499) and the Open Access Publication Fund of the University of Greifswald.

## Additional information

### Funding

| Funder | Grant reference number | Author |
|--------|------------------------|--------|
| Institut für Marine Biotechnologie e.V. | | Tjorven Hinzke<br>Mareike Meister |
| German Academic Exchange Service | | Tjorven Hinzke |
| NC State University | Chancellor's Faculty Excellence Program Cluster on Microbiomes and Complex Microbial Communities | Manuel Kleiner |
| National Institute of Food and Agriculture | Hatch project 1014212 | Manuel Kleiner |
| National Science Foundation | OCE-1131095 | Stefan M Sievert |
| Woods Hole Oceanographic Institution | The WHOI Investment in Science Fund | Stefan M Sievert |
| German Research Foundation | MA6346/2–1 | Stephanie Markert |
| National Science Foundation | OCE-1559198 | Stefan M Sievert |
| German Research Foundation | DFG Open Access Publication Fund: 393148499 | Stephanie Markert |

The funders had no role in study design, data collection and interpretation, or the decision to submit the work for publication.

### Author contributions

Tjorven Hinzke, Data curation, Formal analysis, Validation, Investigation, Visualization, Methodology, Writing - original draft, review, editing; Manuel Kleiner, Supervision, Methodology, Writing - review and editing; Mareike Meister, Christian Hentschker, Petra Hildebrandt, Florian Bonn, Investigation; Rabea Schlüter, Resources, Investigation, Visualization; Jan Pané-Farré, Investigation, Visualization; Horst Felbeck, Supervision, Investigation; Stefan M Sievert, Resources, Funding acquisition, Writing - review and editing; Uwe Völker, Dörte Becher, Resources; Thomas Schweder, Resources, Supervision; Stephanie Markert, Conceptualization, Supervision, Funding acquisition, Investigation, Visualization, Methodology, Project administration, Writing - review and editing

### Author ORCIDs

Tjorven Hinzke (iD) https://orcid.org/0000-0003-1117-0235
Manuel Kleiner (iD) http://orcid.org/0000-0001-6904-0287
Dörte Becher (iD) http://orcid.org/0000-0002-9630-5735
Thomas Schweder (iD) http://orcid.org/0000-0002-7213-3596
Stephanie Markert (iD) https://orcid.org/0000-0003-0923-3305

### Decision letter and Author response

Decision letter https://doi.org/10.7554/eLife.58371.sa1
Author response https://doi.org/10.7554/eLife.58371.sa2

## Additional files

### Supplementary files

• Supplementary file 1. Sampling details for specimens and sample types used in this study. All animals were collected at the Crab Spa vent site in the East Pacific Rise (EPR) Tica area. For proteomic analyses, *Riftia* trophosome homogenate (Hom) was subjected to Histodenz-based density gradient centrifugation, separating symbiont cells according to their sizes. After centrifugation, the gradient

was carefully disassembled into 24 subsamples/fractions (numbered 1 to 24), all of which were analyzed by CARD-FISH to identify those fractions in which the percentage of very small, small, medium-sized and large symbionts cells was highest. These fractions were designated XS, S, M, and L for the respective worm and included in comparative proteomic analyses.

• Supplementary file 2. Symbiont proteins identified in this study, which were included in statistical analyses. Relative abundance of symbiont proteins in fractions XS, S, M, and L in sulfur-rich (S-rich) and sulfur-depleted (S-depl) trophosomes is displayed as edgeR-RLE-corrected spectral count values, which represent average values of four biological replicates (S-rich) and three biological replicates (S-depl). Abundance trends, that is increase or decrease of relative protein abundance across the four gradient fractions is indicated by spark lines (columns „Trend") and by color shades from light green/light gray (lowest protein abundance across all four fractions) to dark green/dark gray (highest abundance). Significant changes in S-rich or S-depl specimens (or both) are indicated by * (these trends are consistent in all replicates according to STEM trend analysis and pairwise comparison between fractions by random forests; for a detailed definition of significance as applied in this study see Materials and methods). Protein accession numbers refer to NCBI/Uniprot entries (EGV- and EGW- accessions) and JGI entries (all other accessions). The prefix 'Sym_' indicates that this accesion number refers to a symbiont protein in our combined host-and-symbiont database, while host proteins have the prefix 'Host_' (note that the prefixes were omitted in *Figure 3* and *Figure 4* in the main text for readability's sake). Please note that this table includes only such identified symbiont proteins, which were detected with at least five spectral counts in a minimum of four (of 16) individual replicate S-rich samples or a minimum of three (of 12) individual S-depl samples in fractions XS - L (see Materials and methods for details). Note also, that edgeR-RLE-corrected spectral count values as displayed here can be used to compare a given protein's abundance between the individual fractions, but do not allow for comparisons between proteins of the same sample. A complete list of all symbiont protein identifications, including low abundant proteins and proteins detected in unenriched homogenate samples, which allows for abundace comparison between proteins is presented in b. (b) Unfiltered list of all symbiont proteins identified in this study in density gradient fractions XS, S, M, and L and in unenriched trophosome tissue homogenate (Hom) from sulfur-rich (S-rich) and sulfur-depleted (S-depl) *Riftia* specimens. Relative protein abundance is displayed as %orgNSAF (normalized spectral counts, see Materials and methods), which give a protein's abundance as percentage of all symbiont proteins in the same sample, allowing for comparison between individual proteins within a given sample. %orgNSAF values are average values of four biological replicates (S-rich) and three biological replicates (S-depl). A protein's 'abundance rank' indicates overall abundance across all samples (rank one being the most abundant protein). The 100 most abundant proteins according to this ranking are highlighted in yellow in column B. Protein accession numbers refer to NCBI/Uniprot entries (EGV- and EGW- accessions) and JGI entries (all other accessions). The prefix 'Sym_' indicates that this accesion number refers to a symbiont protein in our combined host-and-symbiont database, while host proteins have the prefix 'Host_' (note that the prefixes were omitted in *Figure 3* and *Figure 4* in the main text for readability's sake). Proteins in gray font are low-abundant proteins, which were not included in statistical analyses. %orgNSAF values of these low-abundant proteins are less reliable and should be interpreted with care. Proteins that were exclusively identified in homogenate samples were also excluded from statistical analyses and are therefore also set in gray font. Please note that %orgNSAF values cannot be compared accross sample types, due to the unequal total identification numbers in the individual sample types. For cross-sample comparisons, please refer to the edgeR-corrected values in a, which contains all proteins in black font.

• Supplementary file 3. Abundance trends of Endoriftia proteins in various metabolic categories across the four fractions XS to L in sulfur-rich (S-rich) and sulfur-depleted (S-depl) *Riftia* specimens. Trends are indicated by color shades from light green/light gray (lowest protein abundance across all four fractions) to dark green/dark gray (highest abundance across all four fractions; note that colors do not allow comparison of protein abundance between proteins). Abundance values are based on statistical evaluation of four biological replicates (S-rich) and three biological replicates (S-depl). Proteins marked with asterisks show statistically significant trends, that is, differences that are consistent across all replicates in S-rich or S-depl specimens (or both). White cells indicate that this protein was not detected in this sample or was too low abundant to be included in statistical analyses. For

an overview of all identified symbiont proteins and their relative abundances see *Supplementary file 2*. Protein accession numbers refer to NCBI/Uniprot entries (EGV- and EGW-accessions) and JGI entries (all other accessions). The prefix 'Sym_' indicates that this accession number refers to a symbiont protein in our combined host-and-symbiont database (note that the prefix was omitted in *Figure 3* and *Figure 4* in the main text for readability's sake).

• Supplementary file 4. *Riftia* host proteins with significantly higher abundance (FDR 0.05) in fractions XS and S (containing predominantly very small and small symbiont cells, respectively), compared to the non-enriched trophosome homogenate (Hom). Spectral count data from sulfur-rich (S-rich) and S-depleted specimens were normalized separately in edgeR using RLE normalization (RLE-SC). For details regarding statistical analysis see Materials and methods. In the significance columns, '1' refers to significantly higher abundance in fractions XS and S, '0' to non-significant differences. Blast-KOALA functional categories: FE: family eukaryote, GE: genus eukaryote. TH: transmembrane helices. SP: signal peptide. WoLF PSORT subcellular location prediction: cyto: cytosol, cysk: cytoskeleton, E.R.: endoplasmic reticulum, extr: extracellular, golg: golgi apparatus, lyso: lysosome, mito: mitochondrial, nucl: nucleus, pero: peroxisome, plas: plasma membrane. TargetP secretory pathway prediction: M: mitochondrion, S: secretory pathway, '_': other location. Phobius/SignalP signal peptide prediction: Y: signal peptide predicted, N: no signal peptide predicted. Accession numbers refer to the combined host-and-symbiont database used for protein identification in this study (see Materials and methods, *Hinzke et al., 2019*). The prefix 'Host_' indicates that this accession number refers to a host protein (note that the prefix was omitted in *Figure 4* in the main text for readability's sake).

• Supplementary file 5. Total (summed up) relative abundance of Endoriftia proteins involved in specific metabolic categories in fractions XS, S, M, and L in sulfur-rich (S-rich) *Riftia* specimens (average values, n = 4) and sulfur-depleted (S-depl) *Riftia* specimens (average values, n = 3). Only those 1212 symbiont proteins presented in *Supplementary file 2a*, which are included in the edgeR statistical evaluation, are included (proteins with low abundance and/or only one or two replicate values were excluded). To allow comparison and summing of protein abundances across proteins within one sample, edgeR-RLE-corrected spectral count values were normalized (a) to protein size and (b) to the sum of all proteins before summing up the proteins within categories (100% = all proteins in *Supplementary file 2a*). These results indicate that morphological differences between individual symbiont differentiation stages are accompanied by a gradual change in metabolic function. During differentiation from small to large cells, *Riftia* symbionts rearrange their metabolic priorities, allocating resources to those processes that are most important in their respective life phase and role in the symbiosis.

• Supplementary file 6. *Riftia* trophosome homogenate and gradient fractions enriched in small and large symbionts, respectively, were stained with Syto9 and subjected to flow cytometry analysis in a FACSAria high-speed cell sorter with 488 nm excitation (see Materials and methods for details). Two cell populations were identified, Pop1 and Pop2, which correspond to smaller and larger symbiont cells, respectively (see main text *Figure 2* and *Figure 2—figure supplement 1*). Median fluorescence intensity (FI) per particle, a measure of DNA content per cell, was compared between the two populations 1 and 2 to quantify differences in genome copy number between smaller and larger symbionts (column 'ratio'). Note that FI ratios were not calculated for samples consisting of sorted populations (bottom rows), because these samples contained high cell numbers of either of the two populations, but very low cell numbers of the respective other population, preventing meaningful comparison. Analyses were performed with samples from two *Riftia* specimens (two biological replicates, BR).

• Supplementary file 7. Proteins identified as likely involved in dissimilatory sulfur metabolism in *Ca*. E. persephone after Blast-comparison against proteins identified in the literature: *Weissgerber et al., 2014*.; *Rodriguez et al., 2011*; *Gregersen et al., 2011*. Significant - protein abundance significantly different between fractions containing symbionts of different size (see Materials and methods for details on statistical analysis). Y - yes, N - no, M - maybe.

• Transparent reporting form

## Data availability

The mass spectrometry proteomics data have been deposited to the ProteomeXchange Consortium (ProteomeXchange - ProteomeCentral) via the PRIDE partner repository (Vizcaíno et al., 2016) with the dataset identifier PXD016986.

The following dataset was generated:

| Author(s) | Year | Dataset title | Dataset URL | Database and Identifier |
|---|---|---|---|---|
| Hinzke T, Kleiner M, Meister M, Schlüter R, Hentschke C, Pané-Farré J, Hildebrandt P, Felbeck H, Sievert SM, Bonn F, Völker U, Becher D, Schweder T, Markert S | 2020 | Metabolic differences between morphologically distinct symbiont populations in the tubeworm Riftia pachyptila | https://www.ebi.ac.uk/pride/archive/projects/PXD016986 | PRIDE, PXD016986 |

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

## Appendix 1

### Supplementary results and discussion

### A. Symbiotic Endoriftia cells exist in a remarkable size range

Symbiont cell sizes in *Riftia* trophosome tissue range from 1 to 2 μm to more than 15 μm (main text: *Figure 1*, *Figure 5*). This is in line with previous microscopy-based observations, which suggested that the symbiont cells differentiate from small rod-shaped cells in the trophosome lobule center to larger coccoid cells towards the lobule periphery (*Bright and Sorgo, 2003*). With an about 10-fold increase in diameter, Endoriftia cells enlarge their volume by a factor of ~1000 during their development from smallest to largest coccoid symbiont cells. Considerable enlargement of bacterial cells in the course of symbiotic differentiation has also been observed in the intracellular thiotrophic symbiont of the shallow water clam *Codakia orbicularis* (increases 10-fold in length; *Caro et al., 2007*), in *Sinorhizobium meliloti* in alfalfa nodules (increases four- to seven-fold in length; *Oke and Long, 1999*), in symbionts of the nematode *Eubostrichus* (increase up to 13-fold in length; *Pende et al., 2014*), and in the giant bacterium *Epulopiscium fishelsoni*, intestinal symbiont of surgeon fish (increases up to 3,000-fold in volume; *Bresler and Fishelson, 2003*). Such enormous size gradients are rather the exception than the rule in bacteria, however. Cell sizes, that is, length or diameter, of free-living model bacteria like *Bacillus subtilis* or *Escherichia coli* usually vary only by factor 2 (during cell division), that is, mathematically, these bacteria may increase their volume twofold (assuming a cylindrical shape) to eightfold (assuming a spherical shape) at most (*Chien et al., 2012*). This suggests that the remarkably large size range observed for Endoriftia presents a consequence of its symbiotic life style.

### B. Comparative analysis of enriched symbiont fractions from S-rich vs. S-depleted *Riftia* specimens

#### Overview

Our comparative analyses of symbiont-enriched fractions XS to L revealed that in both S-rich and S-depleted samples, protein profiles differed with increasing symbiont cell size (*Appendix 1—figure 1*). Many groups of proteins (e.g. carbohydrate metabolism-related proteins) showed similar trends across size classes in S-rich and S-depleted specimens, even if individual protein abundances differed. Statistical testing for significant differences in protein abundance between S-rich and S-depleted fractions of the same size class returned only very few (edgeR) or no (random forest) hits. This may in part be due to the less effective enrichment of symbionts from S-depleted trophosome tissue homogenate. However, very similar abundance patterns in symbionts from sulfur-rich and sulfur-depleted hosts might also reflect the fact that symbionts are very well buffered against environmental changes (as previously suggested, *Hinzke et al., 2019*) and, therefore, functional differences between symbiont morphotypes in S-rich vs. S-depleted symbionts might largely be negligible. Some of these differences, however, seemed to be specific for the respective energy situation and are outlined below.

#### Cell division

In sulfur-depleted hosts, *Riftia* symbionts appear to divide less frequently than in sulfur-rich specimens, as indicated by lower abundance of the major cell division protein FtsZ in all S-depleted fractions compared to their S-rich counterparts (*Supplementary file 2*; please note that, due to its low abundance, FtsZ was not included in statistical analysis in S-depleted samples). In S-depleted fraction XS, FtsZ abundance was about 3.5 times lower than in S-rich fraction XS.

Less symbiont cell division in S-depleted *Riftia* accords with the idea of severe energy limitation in sulfur-depleted symbionts and is in agreement with our previous finding that symbiont proteinaceous biomass is lower in trophosomes of S-depleted specimens (*Hinzke et al., 2019*). In this previous study, we suggested that S-depleted hosts digest a larger part of their symbiont population as compared to S-rich tubeworms. As the host mainly digests large symbionts at the trophosome lobule periphery (*Figure 5* main text; *Bright and Sorgo, 2003*), one might expect that more digestion leads to relatively more smaller symbionts in S-depleted trophosomes as compared to S-rich hosts.

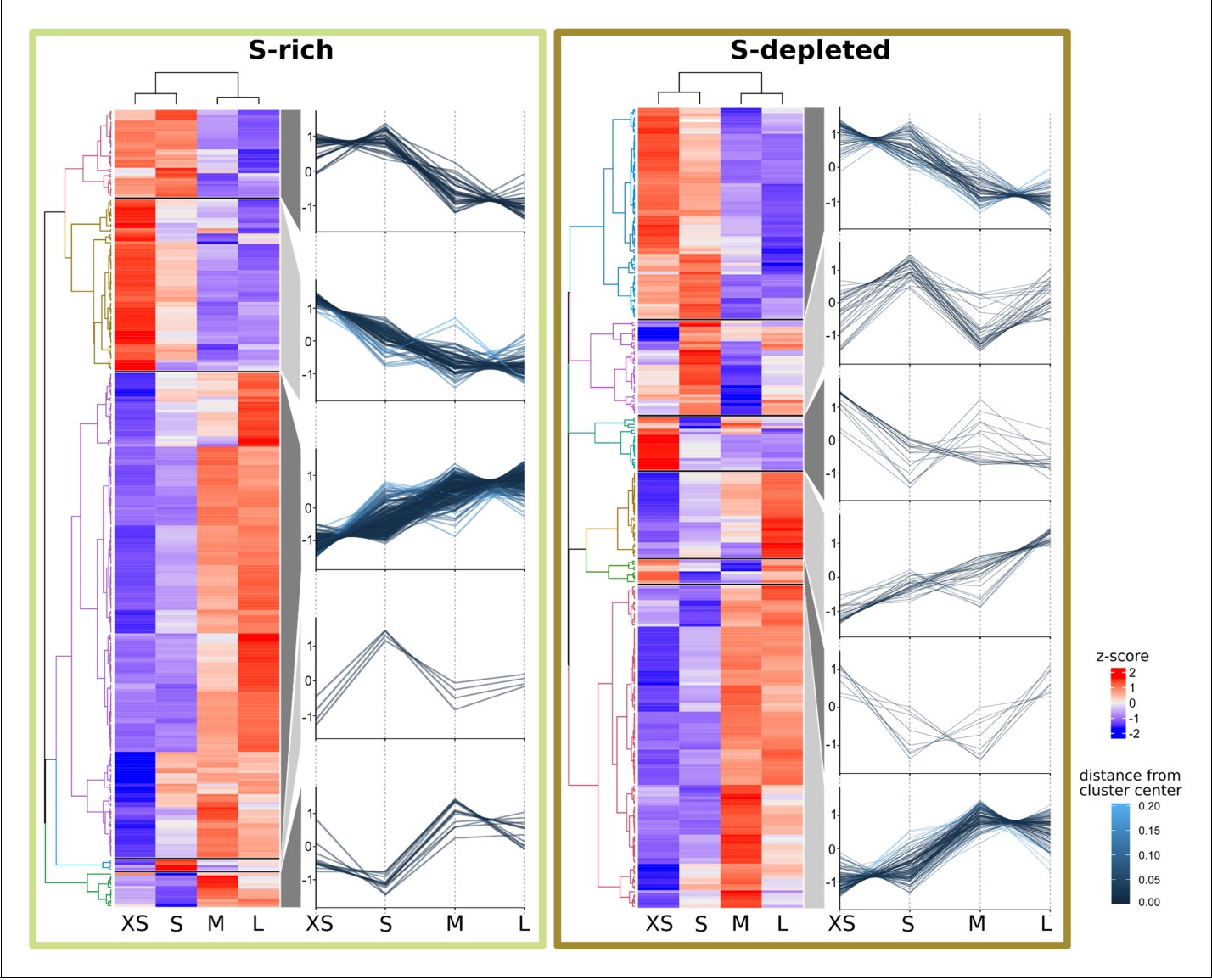

**Appendix 1—figure 1.** Abundance trends of 465 *Riftia* symbiont proteins with significant abundance differences between the four analyzed gradient fractions XS (enriched in very small symbiont cells) to L (containing the highest percentage of large symbiont cells) in S-rich and S-depleted *Riftia* trophosomes. Heat maps show relative protein abundances (z-scores of edgeR-RLE-corrected spectral count values; see Methods for details) and line graphs indicate trends in the observed differences.

This was, however, not the case, as symbiont size distribution was quite comparable in trophosome homogenates of S-rich and S-depleted trophosomes (see *Figure 1*, main text). Such similar distributions would be in line with the hypothesis that the host may digest small symbionts, too, besides large Endoriftia (see main text). However, as we could not see evidence for small symbionts in the process of digestion in any of our TEM images, we rather propose that both more symbiont digestion and less symbiont cell division co-occur in S-depleted worm specimens, leading to the previously observed loss in total symbiont biomass.

## Carbon fixation

Enzymes of the two autotrophic pathways, Calvin cycle and rTCA cycle, did not show significant differences in protein abundance patterns when comparing S-rich and S-depleted gradient fractions. However, based on our SIF analysis (*Appendix 1—figure 2*, see below), both pathways could be differentially used between the two sample sets: Throughout all gradient fractions, S-depleted samples

had a notably more positive $\delta^{13}C$ value than the respective S-rich fractions. This indicates that in S-depleted Endoriftia, the relative contribution of the rTCA cycle to overall carbon fixation is higher than in S-rich symbionts. This is in good agreement with earlier findings of comparative proteomic analyses (*Markert et al., 2007*). Since the rTCA cycle is more energy-efficient than the Calvin cycle, this shift towards a higher contribution of the less costly autotrophic pathway might be beneficial during energy limitation.

## Growth-related processes

Highest abundance of RNA polymerase subunits, transcription elongation factors, transcription anti-termination protein and various translation-related proteins in fraction XS of S-rich and S-depleted specimens indicates that small symbionts devote relatively more energy and resources to protein synthesis than large symbionts (*Supplementary file 3h*). This is in agreement with the idea that small Endoriftia function as actively dividing and growing 'stem cells' of the Endoriftia population, whereas large symbionts have the role of highly efficient biomass producers (see main text).

This proposed greater importance of growth-related processes in small symbionts may result in higher intracellular pyrophoshate levels, as suggested by high abundance of pyrophosphatases in fraction XS. The highly abundant pyrophosphate-energized proton pump HppA (Sym_EGV49909.1) and the inorganic pyrophosphatase Ppa (Sym_EGV49908.1) had their highest abundances in fraction XS in S-rich samples (*Supplementary file 2*). Pyrophosphatases play an important role in energy metabolism by catalyzing the hydrolysis of inorganic pyrophosphates ($PP_i$), which are produced at particularly high rates by biosynthetic reactions in growing cells (*Klemme, 1976*; *Chen et al., 1990*). By removing $PP_i$, pyrophosphatases shift the thermodynamic equilibrium to favor reactions like DNA, RNA, and protein synthesis (*Lahti, 1983*).

HppA may furthermore have an additional growth-related function: During $PP_i$ hydrolysis, HppA pumps protons into the periplasm, thus establishing a proton motive force (*Maeshima, 2000*). As cell division is an energy-expensive process, which requires not only ATP but also proton motive force (*Goehring and Beckwith, 2005*), HppA may be upregulated to accommodate this increased demand in small, dividing Endoriftia. At the same time, HppA presumably increases energy efficiency of the Calvin cycle (*Markert et al., 2011*). Interestingly, HppA abundance was notably lower in S-depleted XS fractions, supporting the idea of reduced cell division in energy-depleted symbionts (see above).

Besides HppA, highly abundant nitrate reductase NarGHI showed a very similar abundance pattern with (i) decreasing abundance from fraction XS to L and (ii) notably lower abundances in S-depleted fractions compared to their respective S-rich counterparts (*Supplementary file 2*). NarGHI, which catalyzes the first step of anaerobic denitrification (see main text), also produces a proton motive force (*Bertero et al., 2003*). We therefore speculate that it may also support cell division in small symbionts, and that its lower abundance in S-depleted samples correlates with less cell division in these symbionts.

## Host interactions

Proteins which may protect the symbiont from digestion by the host could be most important in small symbiont cells and particularly so in S-depleted *Riftia*, as suggested by highest abundances of an ankyrin protein and of the FK506-binding protein FkpA in S-depleted fraction S (*Supplementary file 2*).

In S-rich and S-depleted samples, the ankyrin-like symbiont protein (Sym_EGV51005.1) decreased in abundance from fraction S to L. Endoriftia ankyrin repeat-containing proteins were previously suggested to be involved in microbe-host interactions, possibly to counteract digestion by the host (*Hinzke et al., 2019*). As small Endoriftia are the main dividing symbiont subpopulation and thus ensure survival of the symbiont population as a whole, digesting those cells would harm not only the symbiont, but also the host itself. The ankyrin protein could fulfill a protective role especially for these smaller symbionts.

The *Riftia* symbiont's FK506-binding protein (Sym_EGV50540.1), which showed a comparable abundance trend, might have a similar role. In *Salmonella typhimurium* and *Cronobacter*, FkpA is

involved in survival inside host cells (*Horne et al., 1997*; *Eshwar et al., 2015*), suggesting that the Endoriftia FkpA, too, provides protection for the intracellular symbiont.

## C. Flow cytometry of *Riftia* symbionts

According to our flow cytometry data, *Riftia* trophosome homogenate and enriched gradient fractions were quite heterogeneous (*Figure 2—figure supplement 1*), with a number of other populations present besides populations 1 (small symbionts) and 2 (large symbionts). This heterogeneity may, to a minor degree, result from contaminating host organelles, but is presumably mostly due to the fact that (a) symbionts exist not only as small or large cells, but also adopt any intermediate size, and (b) intracellularly stored sulfur influences the cells' light-scattering properties (especially side scatter, SSC), considerably (as shown for thiotrophic lucinid symbionts; *Caro et al., 2007*).

We sorted one of the additional populations, with SSC between $10^4$ and $10^5$ and FSC between $10^3$ and $10^4$ (i.e. with higher SSC but lower FSC than populations 1 and 2) to examine it separately. Fluorescence microscopy revealed that this population consisted mostly of medium-sized symbionts, which – unlike populations 1 and 2 – contained numerous sulfur globules (images not shown). It could be assumed that other symbiont cell populations, for example, small S-rich and large S-rich cells, might also be present. This hypothesis awaits confirmation in future studies. To estimate symbiont DNA content in the present study, we only included populations 1 and 2, which were readily comparable due to their similar sulfur content (i.e. there were hardly any sulfur globules visible).

As also described for a thiotrophic lucinid symbiont (*Caro et al., 2007*), cell populations were not entirely congruent across the two bioreplicates in our Endoriftia flow cytometry analyses. Consequently, individual fluorescence intensity (FI) values varied by a factor of 2 (on average; *Supplementary file 6*). Nevertheless, both replicates clearly showed the same trend, that is, higher FI per particle in population 2 compared to population 1 across all samples, strongly indicating multiple genome copy numbers in large symbionts.

## D. CO$_2$ metabolism is differentially regulated across Endoriftia cell sizes
### Small and large Endoriftia differ with regard to their autotrophic pathway use

Relative contributions of Calvin cycle and rTCA cycle to autotrophic net carbon fixation appear to differ distinctly between small and large Endoriftia, as revealed by our analysis of stable carbon isotope fingerprints (SIFs, *Appendix 1—figure 2*). The Calvin cycle, with its type II RubisCO, fixes preferentially $^{12}CO_2$ rather than $^{13}CO_2$, that is, it discriminates against $^{13}C$ with a shift of −5 to −25‰. In contrast, the rTCA cycle discriminates only with −2 to −13‰ against $^{13}C$, which makes it possible to distinguish between carbon fixed by either of the two pathways or by varying contributions of both (*Pearson, 2010*). The differences in SIF values we observed point to differential use of Calvin cycle and rTCA cycle in small vs. large symbionts: Fraction XS had the most negative of all $\delta^{13}C$ values, indicating that relatively more of the carbon in these samples was fixed by the Calvin cycle key enzyme RubisCO than in the other fractions, that is, the Calvin cycle's relative contribution to autotrophy is highest in small symbionts. With growing symbiont cell size, this contribution appears to decrease, as indicated by more positive $\delta^{13}C$ values. Highest relative $\delta^{13}C$ values in fractions M and L suggest that the relative contribution of the rTCA cycle to fixed carbon was highest in large symbionts.

In addition to this, we observed that $\delta^{13}C$ values were notably more positive in all S-depleted samples, compared to their S-rich counterparts. This strongly indicates that the rTCA cycle, which is presumably more energy-efficient than the Calvin cycle, is relatively more used in S-depleted (i.e. energy-depleted) symbionts of all sizes (see Appendix section B above).

### The carbon fixation key enzyme RubisCO is more abundant in large Endoriftia

The Calvin cycle key enzyme RubisCO was detected with notably higher mRNA-based fluorescence intensities in large Endoriftia cells, compared to smaller symbionts. This is in agreement with our proteomic results (see main text), and supports the conclusion that large symbionts are more involved in carbon fixation and, generally, in biomass production, than small symbionts

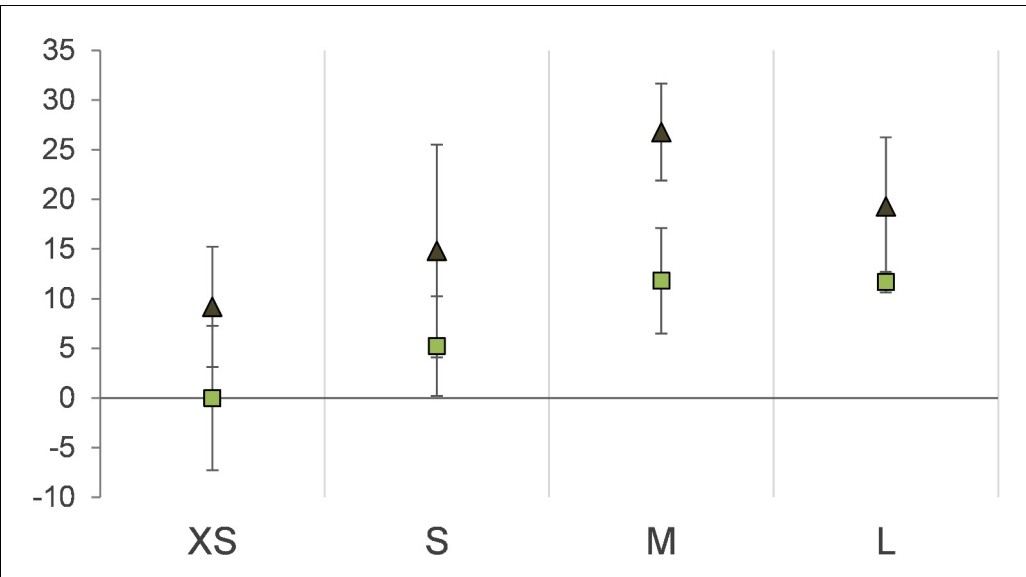

**Appendix 1—figure 2.** Protein stable carbon isotope values ($\delta^{13}$C values) of *Riftia* gradient fractions enriched in symbionts of different cell size (XS – L) relative to fraction XS of S-rich trophosome as baseline. Light squares: S-rich symbionts (average values, n = 4), dark triangles: S-depleted symbionts (n = 3). Error bars indicate standard errors of the mean.

(*Appendix 1—figure 3*). Potential differences in mRNA fluorescence intensity of the rTCA cycle key enzyme ATP-citrate lyase subunit B (AclB) were poorly distinguishable between smaller and larger symbiont cells. However, an AclB signal was quite clearly visible in the very same large cells in which RubisCO was detected. This strongly suggests that Calvin cycle and rTCA cycle are expressed simultaneously by the same symbiont cells.

## Expression patterns of TCA cycle enzymes are ambiguous

Like RubisCO, the rTCA cycle key enzyme ATP-citrate lyase small subunit (AclA, Sym_2601634392) was detected with significantly increasing abundance from fraction XS to L in our proteomic analyses, suggesting that carbon fixation plays a relatively more important role in large *Riftia* symbionts than in small symbionts (see main text). However, expression of other (r)TCA cycle enzymes was surprisingly inconsistent, that is, while abundance of some enzymes increased towards fraction L (including the key enzymes AclA and KorAB), other enzymes showed the opposite, albeit non-significant, trend and became less abundant (e.g. AclB, isocitrate dehydrogenase Icd; *Supplementary file 3d*, *Supplementary file 2*). Further contributing to this ambiguous pattern, the AclB mRNA signal was detected with very similar (and very low) abundances in small and large Endoriftia cells (see *Appendix 1—figure 3*). A possible explanation for these observations might be that Endoriftia's (r)TCA cycle enzymes can run in either direction, depending on cellular requirements. While certain key reactions of TCA and rTCA cycle have long been considered as irreversible, this seems not always to be the case, as, for instance, reported for citrate synthase, key enzyme of the oxidative TCA cycle, which can also operate in the reverse direction, cleaving citrate (*Mall et al., 2018*). Endoriftia's citrate synthase (although encoded in the genome) was not detected at all on the protein level in this study, allowing for the speculation that AclAB might functionally replace citrate synthase in the oxidative version of the TCA cycle by running in reverse, possibly even producing ATP in the process. Assuming that the observed discrepancies in Endoriftia (r)TCA cycle enzyme abundance trends are thus indeed caused by flexible changes in the enzymes' operating directions, Icd could, for example, produce oxaloacetate (e.g. for glutamate synthesis) and NADH in small symbionts, while in large symbionts, Icd might fix $CO_2$ by running in the reverse direction. Further studies are required to solve the regulation of the symbiont (r)TCA cycle. The recently described combination of matrix-assisted laser desorption/ionization mass spectrometry and FISH (metaFISH),

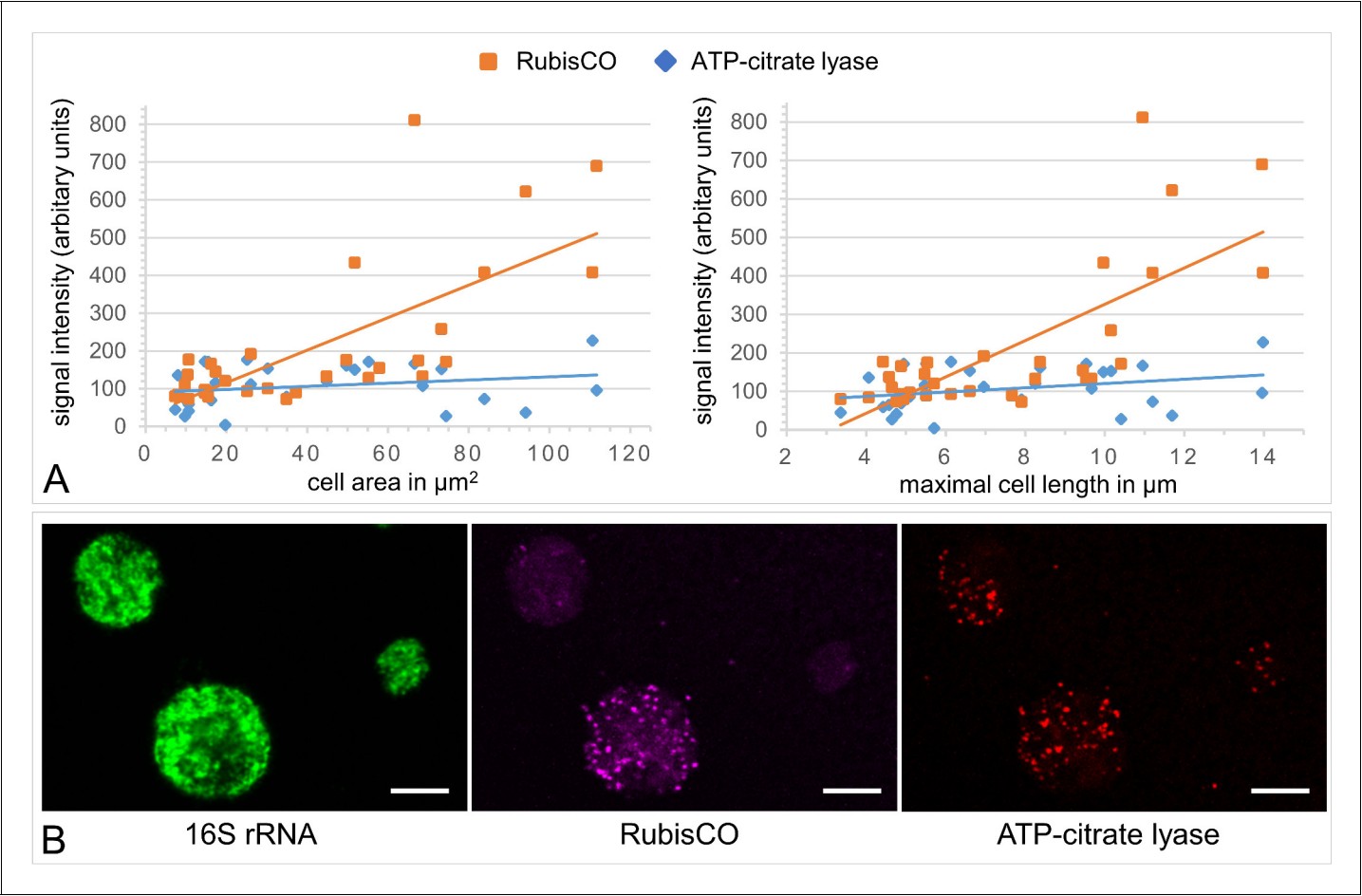

**Appendix 1—figure 3.** A gradient fraction enriched in large symbionts (but also containing small symbiont cells) was fixed as for CARD-FISH analysis and incubated with fluorescently labeled RNA probes against the Endoriftia 16S rRNA and the mRNAs of Calvin cycle key enzyme RubisCO and rTCA cycle key enzyme ATP-citrate lyase (subunit AclB) before examination by confocal laser scanning microscopy (CLSM, see Materials and methods). (**A**) Background-corrected mean signal intensities per pixel calculated from a total of 33 cells (in eight images on a filter from one biological replicate, n = 1) plotted against cell area (left) and Feret's diameter of the cell (right). Straight lines indicate the linear between mean pixel intensities and cell size. Average RubisCO mRNA signal intensity increased notably with cell size (orange lines), while AclB signal intensity increased only very slightly (blue lines). (**B**) CLSM image of Endoriftia cells (the same cells are visible with different fluorophores in the three panels). Supporting the quantitation in (A) and in line with our proteomic results, the RubisCO signal is markedly more intense in large symbiont cells than in small cells, while the AclB signal is very weak and signal intensity differences between large and small cells seem to be minor. Scale bar = 5 µm. Image brightness and contrast were manually adjusted. Hybridizations of several filters from one biological replicate (n = 1) without probes but with fluorophore-carrying hairpins, and without probes or hairpins were used as negative controls and produced no fluorescence signals (images not shown).

which allows for discrimination of symbiont subpopulations based on the metabolites they produce (*Geier et al., 2020*), might be a promising tool for this purpose.

## Large symbionts may take up organic compounds in addition to $CO_2$

Our detection of five *Riftia* symbiont TRAP transporter subunits and four ABC transporter components putatively involved in uptake of organic material with increasing relative abundance from fraction XS to L indicates that Endoriftia imports small organic compounds, particularly in the late stage of differentiation, that is, in large cells. ABC transporters can mediate uptake of small molecules (such as sugars, amino acids, or vitamins), and metal ions (*Davidson et al., 2008*), while TRAP transporters facilitate import of C4-dicarboxylates like fumarate, succinate, and malate (Dct type; *Mulligan et al., 2011*) or of amino acids like glutamate and glutamine (TAXI type; *Mulligan et al., 2007*). All these compounds may be relevant heterotrophic substrates in large Endoriftia, which could channel amino acids and peptides into protein biosynthesis, while sugars could be stored as

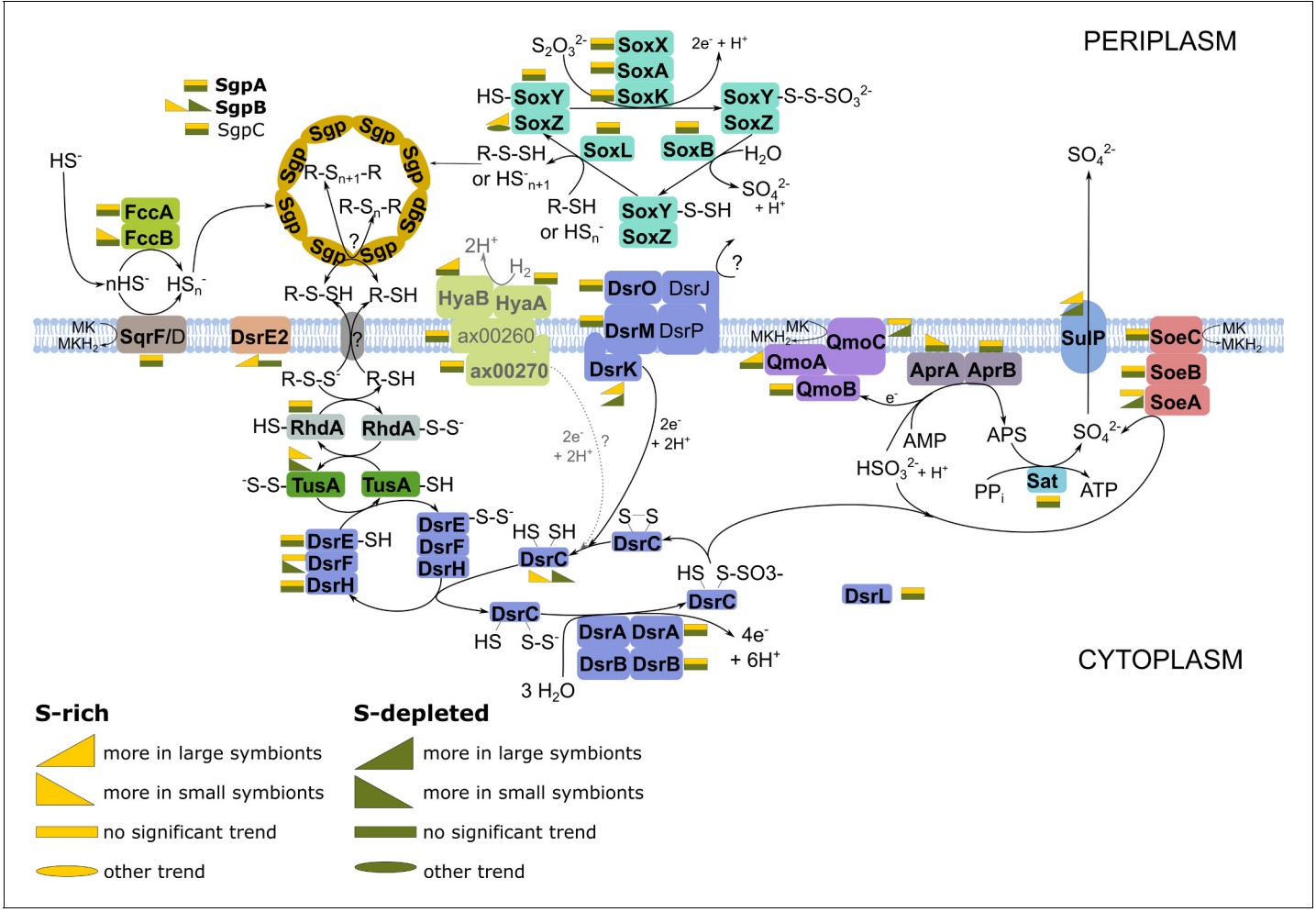

**Appendix 1—figure 4.** Energy-generating oxidation of reduced sulfur compounds in Endoriftia. Proteins in bold were detected in this study. (Figure adapted from *Grein et al., 2010*; *Markert et al., 2011*; *Rodriguez et al., 2011*; *Stewart et al., 2011*; *Dahl et al., 2013*; *Stockdreher et al., 2014*; *Weissgerber et al., 2014*). As the role of hydrogen as electron donor in the *Riftia* symbioses was recently questioned (*Mitchell et al., 2019*), the associated reactions are labeled in gray and with a question mark.

glycogen. Heterotrophy in thiotrophic symbionts was previously shown for a ciliate symbiont (*Seah et al., 2019*) and for ectosymbionts of shrimp (*Ponsard et al., 2013*). Although the *Riftia* symbiont's potential for mixotrophy, that is, for both autotrophy and heterotrophy, had been predicted from the symbiont's genome, it was previously assumed that heterotrophy might be particularly relevant in free-living Endoriftia, but not during symbiosis (*Robidart et al., 2008*). Our results challenge this assumption and suggest that Endoriftia relies on mixotrophy even when in symbiosis, which would allow re-cycling of carbon from host to symbiont.

## E. Small Endoriftia might be nitrogen-limited

Small Endoriftia may rely relatively more on the glutamine synthetase-glutamate synthase (GS-GOGAT) pathway for ammonia assimilation, while large symbionts cells seem to preferably use glutamate dehydrogenase (GDH) for this purpose. In both S-rich and S-depleted samples, a glutamine synthetase copy (GlnA), glutamate synthase subunit GltB and nitrogen regulatory protein P-II (GlnB) were detected with decreasing abundance from fraction XS to L (*Figure 3* main text, *Supplementary file 3f*). In contrast, glutamate dehydrogenase (GdhA) showed the opposite trend with lowest abundance in XS and highest abundance in L (S-rich) or M (S-depleted). The GS-GOGAT pathway, which is energetically more expensive than the GDH-pathway, was shown to be used under energy-rich conditions or during nitrogen limitation in *E. coli* (reviewed in *Reitzer, 2003*). GS-

GOGAT was furthermore shown to have a higher affinity towards ammonium than GDH (*Reitzer, 2003*). This suggests that small symbionts could be nitrogen-limited, either due to a concentration gradient (with highest nitrogen levels in the peripheral lobule zones), and/or due to their own high demand for nitrogen compounds for growth. Further investigations are required to evaluate this speculation.

## F. Sulfur metabolism

While many of the energy-generating reactions of the uncultured *Riftia* symbiont's sulfur metabolism have been elucidated previously (*Markert et al., 2011*), several details remained vague. Our new proteome data enabled us to propose a more detailed model of the *Endoriftia* sulfur metabolism (*Appendix 1—figure 4*, *Supplementary file 7*).

**Appendix 1—table 1.** Nucleotide sequences used for Hybridization chain reaction (HCR) FISH analyses in this study.
See Materials and methods for details.

| >Endoriftia_RubisCO-1–3 | CAACGGGGTAGGCGATCTTCATCAGCTCTTTGGCTTCATCGATCTCATAG |
|---|---|
| >Endoriftia_RubisCO-1–6 | CACATATCCTGGATGTTGACCGCAGGACCGTCGTACAGGCGCAGGTATTT |
| >Endoriftia_RubisCO-1–11 | ATGCACCCAGCAGACGGGTCATCTTGATGTGTACGAAAGCGGTGTAACCA |
| >Endoriftia_RubisCO-1–14 | AAAGGACTCGAAGGCGCGAGCGAACTCTTTGTGCTCTTTCGCGTACTCGA |
| | |
| >Endoriftia_AclB-1–1 | AGACGGCGGTAGATAGACCACCGCCACATTGAATTCACAGCCGTCGTCAA |
| >Endoriftia_AclB-1–6 | CGAACTTCTCCATGAACCACTCTTCCTTGGCGACGGCATTGTCACCGGAA |
| >Endoriftia_AclB-1–8 | CTGGTGTCGGTCGGATCTTCGATGCCTGCTTTCTTGAACAGCTCCATCAT |
| >Endoriftia_AclB-1–12 | GTGGGCGAAACCGGTATGGGTGAGGAAACCGATATAGCCTTTGTTGACCT |
| >Endoriftia_AclB-1–18 | AAACAGGAACGTGGTGAAGGCGGCAGATTCCATGGTCGCGTCGCTGATCT |
| | |
| >Endoriftia_16srRNA-1 | TATTAGCTCGGATTTCTCCGAGTTGTCCCCCACTACTGGGCAGATTCCTA |
| >Endoriftia_16srRNA-5 | ACGGAGTTAGCCGGTGCTTCTTCTAAAGGTAACGTCAAGACCCAAGGGTA |
| >Endoriftia_16srRNA-9 | TTTACGGCGTGGACTACCAGGGTATCTAATCCTGTTTGCTACCCACGCTT |
| >Endoriftia_16srRNA-13 | TCGGCTCCCGAAGGCACCAATCTATCTCTAGAAAGTTCCGAGGATGTCAA |
| >Endoriftia_16srRNA-14 | GTTCCCCTAGGGCTACCTTGTTACGACTTCACCCCAGTCATGAATCACAA |

**Appendix 1—table 2.** Overview of symbiont protein identification numbers in all sample types in this study, that is in gradient fractions XS - L and in non-enriched trophosome homogenate (Hom).
ID count: number of identified proteins. Numbers are based on four biological replicates for sulfur-rich samples and three biological replicates for sulfur-depleted samples. Note that not all proteins were included in statistical analyses (StAn; see Materials and methods for details). GF: gradient fractions.

| | sulfur-rich trophosome | | | | | sulfur-depleted trophosome | | | | | *total* |
|---|---|---|---|---|---|---|---|---|---|---|---|
| | Hom | XS | S | M | L | Hom | XS | S | M | L | |
| | | | | | | | | | | | |
| ID count | 1151 | 1022 | 1296 | 1603 | 1722 | 1017 | 1099 | 1260 | 1605 | 1572 | *1946* |
| ID count (Hom only) | 1151 | | | | | 1017 | | | | | *1223* |
| ID count (total all GF) | | | 1821 | | | | | 1727 | | | *1898* |
| ID count (total all sample types) | | | 1867 | | | | | 1773 | | | *1946* |
| Proteins in StAn | | 940 | 1081 | 1135 | 1134 | | 1008 | 1091 | 1150 | 1143 | *1212* |
| Proteins in StAn (total all GF) | | | 1135 | | | | | 1151 | | | *1212* |

DsrC: The Endoriftia genome encodes several copies of DsrC family proteins, four of which were detected as proteins in this study (*Supplementary file 2*). One of them, Sym_EGV52266.1, was one of the most abundant symbiont proteins, pointing to considerable physiological importance of this protein. Similar to the situation in Endoriftia, three putative DsrC copies were found in the *Calyptogena okutanii* symbiont (*Harada et al., 2009*), and DsrC was also the single most abundant sulfur metabolism mRNA in the *Solemya velum* symbiont (*Stewart et al., 2011*). DsrC has been described to fulfill a key role in dissimilatory sulfur metabolism, including a putative function in transcription regulation and a function as a sulfur trap to allow for maximum DsrAB efficiency (*Venceslau et al., 2014*). Considering this role of DsrC as enhancer of sulfide oxidation efficiency, highest abundance of all Endoriftia DsrC copies in fraction XS (and lowest DsrC abundance in fraction M or L), corroborates our hypothesis of relatively more $H_2S$ oxidation for energy generation in small *Riftia* symbionts (see main text).

SoeABC: In addition to AprAB and SopT, two of the key enzymes of cytoplasmic sulfide oxidation, we also found SoeABC to be expressed in Endoriftia. In *Allochromatium vinosum*, SoeABC catalyzes direct oxidation of sulfite to sulfur, independently of AMP (*Dahl et al., 2013*).

SreABC: We found the putatively sulfur oxidation-related proteins SreABC in the metagenome and detected SreA on the protein level in Endoriftia. While the exact function of SreABC in the oxidation of reduced sulfur compounds is unclear, for *A. vinosum* it was speculated that the Sre proteins could oxidize polysulfides, which are intermediates generated during sulfide oxidation to sulfur (*Weissgerber et al., 2013*).

HyaAB: Endoriftia's uptake hydrogenase HyaAB might be involved in sulfur oxidation. In *A. vinosum*, concentration of the Isp-type hydrogenase HydLS was shown to increase substantially in the presence of sulfide (*Weissgerber et al., 2014*), leading to the proposition that hydrogen-derived electrons may be fed into sulfide oxidation via hydrogenase as illustrated in *Appendix 1—figure 4*. *A. vinosum's* HydL (Alvin_2036) and Endoriftia's HyaB (EGV51840.1) protein sequences are 75.69% identical (NCBI BlastP), indicating that both may have similar functions in sulfur oxidation.

## G. Considerations on the blood flow direction in *Riftia* trophosome lobules

Blood flow across *Riftia* trophosome lobules is commonly accepted to proceed from the lobule periphery inwards to the lobule center, based on observations of *van der Land and Nørrevang, 1977*, *Felbeck and Turner, 1995* and *Bright and Sorgo, 2003* (see also *Figure 6* in the main text). This direction would facilitate a potential substrate gradient with highest concentrations of $CO_2$, $N_2$, $O_2$, reduced sulfur compounds and other substances at the lobule periphery and lowest concentrations in the lobule center. Such a periphery-to-center gradient is in agreement with and corroborates many of our findings. For example, (1) lower oxygen concentrations in the lobule center accord with increasing abundances of all four cytochrome c oxidase subunits (Cco: Sym_EGV51283.1, Sym_EGV51282.1, Sym_2601635240, Sym_EGV51284.1; see *Supplementary file 2*) from fraction XS to M or L. Lowest oxygen levels around the smallest symbionts would furthermore explain highest levels of nitrate-reducing NarGHI in fraction XS. (2) Highest $CO_2$ levels at the lobule periphery support higher abundances of $CO_2$ fixation enzymes in large symbionts, and (3) highest abundances of thiosulfate oxidation proteins in large symbionts may be related to more host sulfide oxidation at the lobule periphery, where sulfide concentrations are highest. Providing highest raw substrate concentrations to the large symbionts would support them in their role as major biomass producers in the symbiosis. Moreover, (4) nutrients released during symbiont digestion in the lobule periphery would likely enter the blood stream and subsequently be transported inwards. Decreasing relative abundances of TRAP-type transporters from fraction L to XS (see section D above) suggest that these compounds are imported by the symbionts along the way, with largest Endoriftia benefiting from highest nutrient concentrations.

*Riftia's* blood flow and ensuing putative substrate gradients thus go hand in hand with symbiont differentiation. Quite possibly, they might even be involved in triggering this differentiation.

