## [Decision Letter]

**Acceptance summary:**

This study demonstrates phenotypic variation among the population of chemosynthetic bacterial symbionts of the giant hydrothermal vent tubeworm *Riftia*, the most iconic example of a deep-sea chemosynthetic symbiosis. The proteomics-enabled work adds new metabolic information parsed by small and large-sized symbionts found in different spatial zones within trophosome lobules.

**Decision letter after peer review:**

Thank you for submitting your article "Bacterial symbiont subpopulations have different roles in a deep-sea symbiosis" for consideration by eLife. Your article has been reviewed by Gisela Storz as the Senior and Reviewing Editor, and two reviewers. The following individual involved in review of your submission has agreed to reveal their identity: Victoria Orphan (Reviewer #3).

The reviewers have discussed the reviews with one another and the Reviewing Editor has drafted this decision to help you prepare a revised submission.

We would like to draw your attention to changes in our revision policy that we have made in response to COVID-19 (https://elifesciences.org/articles/57162). Specifically, we are asking editors to accept without delay manuscripts, like yours, that they judge can stand as eLife papers without additional data, even if they feel that they would make the manuscript stronger. Thus the revisions requested below only address clarity and presentation.

Summary:

Hinzke et al., have managed to separate highly enriched fractions of morphologically distinct symbiont cells from *Riftia pachyptila*, the most iconic example of a deep-sea chemosynthetic symbiosis. It is also worth mentioning that although the symbiosis was discovered in 1981, no one has managed to obtain a pure culture of these symbionts. This study is a major breakthrough in understanding symbiont physiology and interactions with the host. The authors propose that this genetically homogenous symbiont population is structured into distinct functional sub-populations, from a population of smaller “stem cell” symbionts that devote their effort to cell division and surviving host digestion, and larger “workhorse” symbionts that build biomass and synthesize vitamins and cofactors needed for the host's nutrition before being digested intracellularly. This study is based on identification of expressed proteins, an “omics” method, which provides comprehensive information about expression of a diverse range of pathways. For example, they revealed that smaller symbionts appear to express the first step of denitrification, the larger symbionts the subsequent steps. Although the reason for this is not well understood, this is an intriguing result. Going beyond omics, their results are consistent with some previous measurements of symbiont activity under some conditions, in particular carbon fixation, which showed that the larger symbionts incorporate radiolabeled inorganic carbon more rapidly than the smaller symbionts. Cell division is also more often observed in electron micrographs of the smaller symbionts than the larger symbionts, in this study and in others. This is therefore an exciting and reliable study!

The reviewers have a number of comments for improving the paper:

1) There are 2 main research lines included in the manuscript- one focused on symbiont size and associated metabolic variation and the other with a comparison of worms with “sulfur rich” or sulfur poor (unhealthy?) trophosome. Most of the paper emphasizes comparisons between large and small symbionts- indeed nearly all of the Discussion focuses on this, while the differences between S-rich and S-poor worms are included in the figures, but not really discussed and I think distracts from the current focus. The authors may want to consider writing up the sulfur rich/poor comparison a separate manuscript (or if there are good reasons to keep the datasets together, the authors should provide more context about what S rich/ poor difference means for the physiology of the worm).

2) More context from the literature about symbiont/host physiology is needed as part of the Introduction and/or Discussion to help the non-specialist interpret your findings and conclusions. For example, it's important to provide context about the delivery of substrates to the symbionts through modified hemoglobin and the mode of nutrient acquisition by the host through digestion rather than milking their symbionts. Note that this occurs in the region that overlaps in part where the “large” symbionts are located and how this might impact the findings. The paper could be strengthened by discussing prior physiological experiments that support the semi-quantitative proteomics data here- for example, without context of prior experiments demonstrating that denitrification was been measured in *riftia* trophosome (e.g. citing experiments done by some of the co-authors on this work Hentschel and Felbeck, 1983, etc).

3) An overview figure that summarizes the current understanding of blood flow, symbiont location, symbiont digestion in addition to the TEM mosaic (again this would help readers not familiar with *riftia* ultrastructure/ physiology) would be extremely helpful. Also, a figure or table that summarizes the main metabolic/ physiological differences highlighted in the paper for large and small symbionts.

4) General thought (expanding on the comment in subsection “Metabolic diversity among symbiont size classes”): It is interesting that the symbiont cells that are ultimately digested are large, have multiple genome copies and enhanced rates of CO2 fixation. Perhaps investing more cellular resources in N-rich biomolecules rather than more refractile lipids prior to digestion provides a greater nutritional advantage the host? Also, I imagine a shift from S storage to low S content in the large symbionts would also aid in digestion. It would be great to see a little more discussion/ interpretation from the host side and the symbiosis rather than focusing primarily on differences in symbiont metabolism.

5) Two key drivers of gene expression in larger symbionts are postulated - relaxation of control of gene regulation, and adaptation to distinct microenvironments. These seem to contradict each other. The shift to a different metabolism tends to point to adaptation rather than relaxed control, which would simply mean that those pathways expressed by the small symbionts, plus many others, are expressed by the large symbionts. This should be clarified.

6) Link from hydrogen oxidation to sulfur metabolism - this is shown as electron flow to H2 to DsrC in Appendix—figure 4, but there was not anything that would directly link the hydrogenase to DsrC in the relevant references (Mitchell, Weissgerber). Please either explain why the link is shown so directly between these two systems, or end the arrows from the hydrogenase with a question mark if this is unclear.

7) The terms “stem cell” and “differentiation” are regularly applied to bacterial symbiont cells in *Riftia* and in plant nodules, but these strictly only apply to processes known from eukaryote cells. If you would like to keep these terms, perhaps put them in inverted commas to reflect this?

8) HCR-FISH can be multiplexed, different fluorophores were used for each target in this study. This MS still does not address the question (but the methods used could!) of whether the same symbiont cell expresses the Calvin cycle and rTCA cycle simultaneously. A negative control for the HCR-FISH experiments is also missing - a no-probe control at the minimum would be required. A nonsense probe (e.g. reverse complement) would also be important.

9) There is a reverse trend in the sulfur-starved animals with respect to RuBisCO expression in small vs large symbiont cells. Small symbiont cells look like they express more RuBisCO when the animal is generally experiencing energy limitation. This seems to contradict one of your major conclusions, this should be clarified.

10) You propose that partitioning metabolic roles into different symbiont sub-populations increases overall productivity. I have to admit that although this theory is cited often in symbiosis, I don't fully understand it. Also, here, what exactly would be the benefits of such partitioning? How would partitioning cell division, host interaction, sulfur oxidation and carbon fixation increase overall productivity? Outlining the arguments for this would help. See a new paper in the ISME Journal (Ankrah et al., for some thoughts about this). Alternatively, you raise the possibility that these distinct metabolic functions reflect adaptation to microenvironments with differing availability of substrates in the worm's trophosome. The current data doesn't really allow you to identify which is the controlling factor here.

[Editors' note: further revisions were suggested prior to acceptance, as described below.]

Thank you for resubmitting your work entitled "Bacterial symbiont subpopulations have different roles in a deep-sea symbiosis" for further consideration by eLife. Your revised article has been evaluated by the original reviewers and Gisela Storz as the Senior and Reviewing Editor.

The manuscript has been improved. The reviewers appreciated your detailed response to all questions and suggestions. They felt the restructuring of the paper, focusing on the physiological heterogeneity in different size classes, and shifting most reporting and discussion on differences between S-rich and S-poor samples has improved the manuscript. The addition of the relative stable carbon isotope values of proteins also strengthened the study. Additionally, the new schematic (Figure 6) is well done and a great addition to the manuscript.

However, there are some remaining issues that need to be addressed before acceptance, as outlined below:

1) There were still questions about the HCR-FISH results. Results should be placed in an appendix or supplement if they are less relevant to the core message of the paper, not if they are questionable, which seems to be implied by the response to point 8 (“did not yield satisfying results so far”). The same standards of robustness and reliability should apply to results in the main text and the supplement. We do not agree that a control for non-specific detection of targets (NSD according to the nomenclature of Nikolakakis et al., 2015) is optional, if a justification based on RNA hybridization behavior is given and can be backed up by literature. “Tremendous costs” also is a weak argument for not including a particular negative control.

---

## [Author Response]

The reviewers have a number of comments for improving the paper:1) There are 2 main research lines included in the manuscript- one focused on symbiont size and associated metabolic variation and the other with a comparison of worms with “sulfur rich” or sulfur poor (unhealthy?) trophosome. Most of the paper emphasizes comparisons between large and small symbionts- indeed nearly all of the discussion focuses on this, while the differences between S-rich and S-poor worms are included in the figures, but not really discussed and I think distracts from the current focus. The authors may want to consider writing up the sulfur rich/poor comparison a separate manuscript (or if there are good reasons to keep the datasets together, the authors should provide more context about what S rich/ poor difference means for the physiology of the worm).

Indeed, we pursued two interconnected research lines, with (i) the comparison of small and large symbionts being the main focus of the study, and (ii) the analysis of how cell size-specific features may vary depending on the energetic situation of the symbiosis (in S-rich and S-depleted trophosomes) as an additional question of interest. Sulfur-depletion in the trophosome is not an indicator of unhealthiness, but merely shows that these symbionts had less stored sulfur and were therefore probably energy-limited (unlike in the S-rich worms, whose symbionts were not energy-limited). Since the symbionts are uncultured, the influence of the prevailing micro-energy regime (S content) on, for example, cell size-dependent metabolic differences in symbiont subpopulations can only be investigated by comparing naturally occurring S-rich and S-depleted trophosomes. Interestingly, throughout all gradient fractions in our study, only very few Endoriftia proteins showed significantly different abundances between S-rich and S-depleted samples. S-rich and S-depleted datasets thus support each other. In light of this mutual support (S-depleted samples functioning as “quasi-replicates” for the S-rich samples), we kept both datasets in the manuscript, but thoroughly adjusted the text and figures for enhanced clarity. We better explained this rationale in the manuscript and extended the definition of “S-rich” and “S-depleted” (Introduction, Materials and methods). Nevertheless, we agree with the reviewer that the two lines of research might be confusing, and the S-rich vs S-depl comparison may distract from the main focus of the study. In the heat map in Figure 3, we therefore now only compare protein abundances across gradient fractions XS – L of S-rich samples. The previous Figure 3 with S-rich and S-depleted samples was moved to Figure 3—figure supplement 1. Moreover, we removed the “S-depl” columns in Figure 4 altogether. In the results, we added a sentence explaining that sulfur-rich and sulfur-depleted samples showed overall quite similar symbiont cell size distributions and protein abundance patterns. “The following results will therefore not discriminate between both sample types and subsequent figures will focus on S-rich samples (unless otherwise stated)” (subsection “Protein identifications and relative protein abundance”).

2) More context from the literature about symbiont/host physiology is needed as part of the introduction and/or discussion to help the non-specialist interpret your findings and conclusions. For example, it's important to provide context about the delivery of substrates to the symbionts through modified hemoglobin and the mode of nutrient acquisition by the host through digestion rather than milking their symbionts.

We agree that this would be helpful and added a paragraph with additional general background information on the physiology of the symbiosis, including nutrient transfer from symbionts to host and substrate supply from host to symbionts as suggested (Introduction).

Note that this occurs in the region that overlaps in part where the “large” symbionts are located and how this might impact the findings.

The experimental approach we used – density gradient centrifugation – does not support the enrichment of symbiont cells in the state of being digested (labelled as S* in Figure 5). This is because only intact symbiont cells remained after homogenization of the tissue (while host cells were broken) and only these intact symbiont cells would migrate through the gradient, while debris from broken cells (including half-digested cells) – accumulates at the very top of the gradient, which we did not analyze (e.g., proteins, membrane fractions). Although our “large symbionts” are in situ located closer to the digestive trophosome lobule zone than the smallest symbionts, we did not observe elevated levels of stress proteins or other signs that would indicate that the physiology of large symbionts is in any way impacted by imminent digestion (as stated in the Discussion).

The paper could be strengthened by discussing prior physiological experiments that support the semi-quantitative proteomics data here- for example, without context of prior experiments demonstrating that denitrification was been measured in riftia trophosome (e.g. citing experiments done by some of the co-authors on this work Hentschel and Felbeck, 1983, etc).

Thank you for these suggestions. We added references for experimental evidence of nitrate respiration in *Riftia* (subsection “Hydrogen oxidation is more relevant in large symbionts”), for the function of host carbonic anhydrase (subsection “Interaction-specific host proteins”), for sulfur storage in globules (subsection “Small Endoriftia store more sulfur and are more involved in sulfide oxidation”), and for sulfide and thiosulfate use in Endoriftia (subsection “In large symbionts, thiosulfate oxidation plays a more prominent role”) in the Discussion. Unfortunately, dedicated literature about the individual physiological properties of smaller and larger *Riftia* symbionts, which we could summon in support of our results, is rather limited. Nevertheless, a broader background about the physiology of the symbiosis is certainly helpful for the reader, which is why we added additional general information and references, including most of those suggested by the reviewers at the end of this letter, in the Introduction. We also cite our comprehensive metaproteomic analysis of host and symbiont physiology, which was published previously (Hinzke et al., 2019).

3) An overview figure that summarizes the current understanding of blood flow, symbiont location, symbiont digestion in addition to the TEM mosaic (again this would help readers not familiar with riftia ultrastructure/ physiology) would be extremely helpful. Also, a figure or table that summarizes the main metabolic/ physiological differences highlighted in the paper for large and small symbionts.

We seized the reviewer’s suggestion and included a new figure (Figure 6) which illustrates where the symbionts are located (in the worm and in the tissue lobules), where the smaller and larger symbionts are and where the latter are digested, and in which direction the host’s blood flows. To summarize the essential findings of our study, we also included the small and large symbionts’ most distinguishing features that determine their respective roles in the symbiosis in the figure.

4) General thought (expanding on the comment in subsection “Metabolic diversity among symbiont size classes”): It is interesting that the symbiont cells that are ultimately digested are large, have multiple genome copies and enhanced rates of CO2 fixation. Perhaps investing more cellular resources in N-rich biomolecules rather than more refractile lipids prior to digestion provides a greater nutritional advantage the host? Also, I imagine a shift from S storage to low S content in the large symbionts would also aid in digestion. It would be great to see a little more discussion/ interpretation from the host side and the symbiosis rather than focusing primarily on differences in symbiont metabolism.

Thank you for these suggestions. We added a sentence about the advantage of multiple genome copies (N-rich biomolecules, subsection “Metabolic diversity among symbiont size classes”), and one regarding the benefit of less sulfur in large symbionts for host-mediated digestion in the Discussion. We also elaborate some more on the host’s side of the symbiosis in the introduction and on the advantages of labor division for the holobiont in the discussion. This should provide a more comprehensive picture of the entire symbiosis to the reader and complements our research focus, i.e., the comparative analysis of differently sized symbiont cells and their physiology.

5) Two key drivers of gene expression in larger symbionts are postulated - relaxation of control of gene regulation, and adaptation to distinct microenvironments. These seem to contradict each other. The shift to a different metabolism tends to point to adaptation rather than relaxed control, which would simply mean that those pathways expressed by the small symbionts, plus many others, are expressed by the large symbionts. This should be clarified.

Maybe this was a bit misleading: Rather than considering “less stringent control” and “adaptation to micro-conditions caused by a substrate gradient” as two different key drivers, we believe that relaxed control in larger symbionts is an adaptation as well. Large symbionts adapt to their specific micro-environment – in which all substrates are plentifully provided and cells are not notably restricted by host interference – by expressing much of the metabolic repertoire simultaneously. We clarified this in the Discussion.

6) Link from hydrogen oxidation to sulfur metabolism - this is shown as electron flow to H2 to DsrC in Appendix—figure 4, but there was not anything that would directly link the hydrogenase to DsrC in the relevant references (Mitchell, Weissgerber). Please either explain why the link is shown so directly between these two systems, or end the arrows from the hydrogenase with a question mark if this is unclear.

The figure was inspired by Weissgerber et al., who suggested this link between hydrogenase and DsrC in their model of sulfur oxidation in *Allochromatium vinosum* (their Figure 3), and labelled it with a question mark to illustrate that it is as yet hypothetical. We indicated the hypothetical character of this connection using a dotted grey line in Appendix—figure 4 and mention in the figure legend that the role of hydrogen in sulfur oxidation in *Riftia* is questionable. To make it even more visible that this link is not based on experimental evidence, we now added a question mark to the arrow in Appendix—figure 4.

7) The terms “stem cell” and “differentiation” are regularly applied to bacterial symbiont cells in Riftia and in plant nodules, but these strictly only apply to processes known from eukaryote cells. If you would like to keep these terms, perhaps put them in inverted commas to reflect this?

We understand the reviewers concern, but would like to keep the terms “stem cells” and “differentiation”. To acknowledge that they were not originally used for bacteria, we put “stem cells” in inverted commas throughout the manuscript and added a short statement explaining our use of the word “differentiation” in the Discussion. “Differentiation” is commonly used to describe the varying morphologies of, for example, *Rhizobia* during their different infection stages in legume root nodules (Van de Velde et al., 2010, Kondorosi et al., 2013) or of heterocyst-forming cyanobacteria (Flores and Herrero, 2010). We would therefore like to adopt the term – in its less strict sense – also for the morphological transformations we see in Endoriftia.

8) HCR-FISH can be multiplexed, different fluorophores were used for each target in this study. This MS still does not address the question (but the methods used could!) of whether the same symbiont cell expresses the Calvin cycle and rTCA cycle simultaneously.

The reviewer is right, thank you for pointing this out. The HCR-FISH analyses were conducted to address exactly the question whether or not RubisCO and the rTCA key enzyme AclB are expressed in the same symbiont cells or if they are expressed by distinct subpopulations in separate trophosome lobule zones. Unfortunately, while the method worked well on isolated symbiont cells, our in situ analyses in trophosome tissue sections did not yield satisfying results so far, preventing an answer to the second part of the question. (We therefore included the HCR-FISH images in the Supplementary Results rather than in the main text.) However, we do see that both enzymes are expressed in the same cells in Appendix—figure 3 (all three panels showing the same symbiont cells with different probes). We made sure to stress this result more clearly in our manuscript (Discussion, Appendix (Supplementary Results and Discussion)).

A negative control for the HCR-FISH experiments is also missing - a no-probe control at the minimum would be required. A nonsense probe (e.g. reverse complement) would also be important.

Two kinds of negative controls were used: (i) Some of the filters were incubated without probes, but with hairpins (to exclude unspecific binding of the fluorophore-carrying hairpin to the tissue), and (ii) some filters were incubated without probes and without hairpins to test for auto fluorescence. In both cases, all negative control images were empty. We added a statement explaining this in the Materials and methods and in the figure caption of Appendix-figure 3. Given the high target-specificity (and tremendous costs) of HCR-FISH probes, a nonsense probe was not considered necessary. There is only one target sequence for RubisCO and AclB in the *Riftia* symbiont genome, and hybridization conditions for RNA probes are so strict that we can exclude unspecific binding to other RNAs or DNA.

9) There is a reverse trend in the sulfur-starved animals with respect to RuBisCO expression in small vs large symbiont cells. Small symbiont cells look like they express more RuBisCO when the animal is generally experiencing energy limitation. This seems to contradict one of your major conclusions, this should be clarified.

Protein abundance values for RubisCO from S-depleted animals are indeed ambiguous. While CbbM abundance increases significantly (i.e., consistently throughout all replicates) from small to large cells in S-rich samples, the trend in the S-depleted samples is not significant (i.e., replicates were quite variable) and abundance differences are relatively small in S-depleted samples. There is no clear trend visible. Possibly – as indicated by our new stable isotope fingerprinting (SIF) analyses (Supplementary Results and Discussion) and by previous results (Markert et al., 2007) – this could be because the Calvin cycle seems to become less relevant in S-depleted (energy-limited) trophosomes, whereas the less “expensive” reductive TCA cycle plays a relatively more important role in symbionts in S-depleted samples. We moved the heat map showing the S-depleted data from Figure 3 to the new Figure 3—figure supplement 1.

10) You propose that partitioning metabolic roles into different symbiont sub-populations increases overall productivity. I have to admit that although this theory is cited often in symbiosis, I don't fully understand it. Also, here, what exactly would be the benefits of such partitioning? How would partitioning cell division, host interaction, sulfur oxidation and carbon fixation increase overall productivity? Outlining the arguments for this would help. See a new paper in the ISME Journal (Ankrah et al. for some thoughts about this). Alternatively, you raise the possibility that these distinct metabolic functions reflect adaptation to microenvironments with differing availability of substrates in the worm's trophosome. The current data doesn't really allow you to identify which is the controlling factor here.

Like in multicellular organisms, dividing responsibilities between dedicated cell populations increases the efficiency of the “holo-organism”. In case of Endoriftia, this specialization (differentiation) allows the symbiont to benefit from both, (i) multiple genome copies with resulting increased transcription efficiency and productivity in large cells, and (ii) high cell division rates in small cells, while at the same time circumventing the individual disadvantages of both stages. These being that large Endoriftia would need to invest much more resources in cell division and replication of their multiple genome copies than small cells, while the smaller cells would not be able to produce as much biomass as the large ones. We included this argumentation in the Discussion. Putative substrate gradients across the trophosome lobules could be one of the drivers of this advantageous specialization, in addition to host-derived influences or effectors. Indeed, we cannot distinguish between potentially controlling factors at this point. Instead, we suggest that a combination of both, the evolutionary advantage of specialization, and different microenvironments, have favored the symbionts’ observed physiological diversity.

[Editors' note: further revisions were suggested prior to acceptance, as described below.]

The manuscript has been improved. The reviewers appreciated your detailed response to all questions and suggestions. They felt the restructuring of the paper, focusing on the physiological heterogeneity in different size classes, and shifting most reporting and discussion on differences between S-rich and S-poor samples has improved the manuscript. The addition of the relative stable carbon isotope values of proteins also strengthened the study. Additionally, the new schematic (Figure 6) is well done and a great addition to the manuscript.However, there are some remaining issues that need to be addressed before acceptance, as outlined below:1) There were still questions about the HCR-FISH results. Results should be placed in an appendix or supplement if they are less relevant to the core message of the paper, not if they are questionable, which seems to be implied by the response to point 8 (“did not yield satisfying results so far”). The same standards of robustness and reliability should apply to results in the main text and the supplement.

The HCR-FISH results presented here are not questionable at all – our apologies if the previous comment was a bit misleading. The results clearly show that both enzymes, RubisCO and AclB, are expressed in the same cells of different sizes in a symbiont cell suspension (specifically: in a gradient fraction enriched in larger Endoriftia after centrifugation of homogenized trophosome tissue).

(What “did not yield satisfying results so far” were hybridizations of the same HCR-FISH probes in trophosome tissue sections or pieces, where tissue autofluorescence posed a considerable difficulty. These are, however, not included or referred to in any part of this manuscript.)

We do not agree that a control for non-specific detection of targets (NSD according to the nomenclature of Nikolakakis et al., 2015) is optional, if a justification based on RNA hybridization behavior is given and can be backed up by literature. “Tremendous costs” also is a weak argument for not including a particular negative control.

We followed the protocol of Nikolakakis et al., 2015 for our experimental design. In their paper on HCR-FISH as a powerful tool in symbiosis research, these authors specify three kinds of controls that should be performed to “identify possible artifacts and confounding effects”: (a) without probes or hairpins, (b) without probes but with hairpins, and – if possible – (c) a “Non-Specific Detection of targets (NSD) control” without the target RNA, but with probes and hairpins.

We followed this recommendation and performed negative control preparations of the (a) and (b) type. A more detailed description of our negative controls in reference to Nikolakakis et al., (2015) was added in the methods section. As Nikolakakis et al., state, an NSD control would only be applicable in two cases, namely (i) where a knockout version or wildtype missing the target is available, or (ii) when the locus of expression of a non-ubiquitous endogenous RNA is known beforehand (in which case the surrounding tissue can be regarded as target-free NSD control). Since Endoriftia is not cultivable (as yet), preventing the generation of knockout strains, and since we did not know the locus of RNA expression in the tissue or cells beforehand, a target-free Endoriftia control sample was, regrettably, unavailable. We did, therefore, take great care to ensure specificity of our HRC-FISH experiment even in the absence of a target-free control sample.

In nucleic acid hybridizations, the hybridization rate approaches zero when mismatches increase to more than 30% (Wetmur, 1991). Consequently, only rather similar target sequences could (theoretically) elicit an “unspecific” binding event, while random binding events to non-similar sequences are excluded based on RNA hybridization behavior. Unlike in other symbioses or in environmental samples, there is only one symbiont phylotype in *Riftia* (Polzin et al., 2019). This considerably reduces the complexity of our sample and eliminates the risk of unspecific binding to RNAs from bacteria other than Endoriftia. Finally, false-positive matches to host RNAs can be excluded for the RubisCO and AclB probes in our images, as symbiont cells were co-labeled with a 16S rRNA-specific probe which confirmed the identity of the cells.

To ensure probe sequence specificity, we searched the *Riftia* symbiont genome assemblies (NCBI whole-genome shotgun projects) NZ_AFOC00000000.1 and NZ_AFZB00000000.1 for matches to our AclB and RubisCO mRNA probe sequences before we started our experiments. Using the selected probe sequences as queries in a BLAST search with parameters set at a low stringency level (BLASTn optimized for somewhat similar sequences, seed length: 7, expectation threshold: 1), several potential off-target hits were identified (see Author response image 1). However, all of these off-target matches were characterized by short sequence coverage (usually between 15 to 30 bp of the total 50 bp probe length) and several mismatches. To test whether the off-target matches could still lead to false-positive detections in our experiment, we performed *in silico* analyses of their hybridization behavior using mathFISH, an online tool for evaluation of oligonucleotide probes for FISH (Yilmaz et al., 2011), at 25% formamide, 45°C, 1 M sodium, and 2 nM probe concentration. As a result of these calculations, all off-target matches yielded hybridization efficiencies of 0%, while all probe sequences produced 100% hybridization efficiency.

**Author response image 1. respfig1:** Blast results of HCR-FISH probes and *in silico* analysis of their hybridization behaviour. Four RubisCO-specific probes and five AclB probes were searched against the NCBI whole-genome shotgun (WGS) sequences NZ_AFOC00000000.1 and NZ_AFZB00000000.1, containing the complete genomes of the *Riftia pachyptila* symbiont and the *Tevnia jerochonana* symbiont, respectively (both symbionts belong to the same 16S rRNA phylotype; Gardebrecht *et al* ., 2012). At low stringency settings (BLASTn optimized for “somewhat similar sequences”, seed length: 7, expectation threshold: 1), most probe sequences (queries) had matches (subjects) not only in their respective target genes, but also produced off-target matches of lower % identity (%Id) and with higher numbers of gaps (g) and mismatches (m) in the alignment. All subject sequences were subjected to *in silico* analyses using the online tool mathFISH at simulated hybridization conditions matching those in our experiments (25% formamide, 45°C, 2 nM probe concentration). As a result, 100% hybridization efficiency was predicted for probe binding to the respective target sequences, whereas binding to off-target matches was calculated with 0% hybridization efficiency. ql: query length; qu start: start of alignment in query; qu end: end of alignment in query; contig numbers beginning with “TevJSym” refer to the the *Tevnia* symbiont genome assembly, contig numbers beginning with “Rifp1Sym” refer to the *Riftia* symbiont genome assembly; CDS: complete coding sequence from start to end; product: protein encoded by coding sequence; HybEff CDS: hybridization efficiency of the probe/target pair (probe = query sequence; target = complete CDS of subject).

Since sequence-based binding to a non-target (but similar) RNA in the samples can thus basically be ruled out, non-specific detection could theoretically only be caused by completely random binding to a non-related structure in the cells. However, in such an event, we would expect that all probes produce a similar signal distribution, independent of their target RNA, which was not the case.

We added a summary of these considerations in the Materials and methods section.